# LOCAL AND UNBALANCED OPTIMAL TRANSPORT FOR FEATURE LEARNING WITH PROBABILISTIC GUARANTEES

## ABSTRACT

This paper explores the local and unbalanced optimal transport for feature learning in an embedding space. Instead of using joint distributions of data, we introduce conditional distributions in terms of the Kullback-Leibler (KL) divergence where some reference conditional distributions are utilized. Using conditional distributions provides the flexibility in controlling the transferring range of given data points. When the block coordinate descent method is employed to solve our model, it is interesting to find that conditional and marginal distributions have closed-form solutions. Moreover, the use of conditional distributions facilitates the derivation of the generalization bound of our model via the Rademacher complexity, which characterizes its convergence speed in terms of the number of samples. By optimizing the anchors (centroids) defined in the model, we also employ the unbalanced optimal transport and autoencoders to explore an embedding space of samples in the clustering problem. In the experimental part, we demonstrate that the proposed model achieves promising performance on some learning tasks. Moreover, we construct a local and unbalanced optimal transport classifier to classify set-valued objects.

## 1 INTRODUCTION

Achieving effective features of data (Boroujeni et al., 2018; Wang et al., 2019; Qian et al., 2019) is a fundamental task in data analysis, and feature learning has been explored in some fields such as machine learning and computer vision. Feature learning aims at exploring a linear or nonlinear transformation to map the original features into an embedding space by optimizing the defined objective function. In the latent representation space, data can be explored, thereby providing some benefits from various learning tasks (Su & Hua, 2019; Sheng & Yuan, 2023).

Earlier feature learning algorithms focus on how to develop effective handcrafted extractors for visualizing high-dimensional data and reducing the effect of the curse of dimensionality. Marginal Fisher analysis (Yan et al., 2007) adopts a graph embedding framework to provide an intrinsic graph with intra-class compactness and a penalty graph with inter-class separability. Max-min distance analysis (Bian & Tao, 2011) achieves the low-dimensional data by maximizing the minimum pairwise distance. A robust linear discriminant analysis method based on the $L_{2,1}$ norm (Nie et al., 2021) is developed to achieve robust projection features, and an effective iterative optimization algorithm is derived to solve a general ratio minimization problem. In (Flamary et al., 2018), Wasserstein discriminant analysis from optimal transport (Cuturi, 2013; Serrurier et al., 2021) is implemented by employing the Wasserstein distance to capture the global and local interactions between classes.

Kernel-based methods that capture the nonlinear features of data have been developed to search for an effective feature space by selecting proper kernel functions. Unlike classical feature learning methods, the embedding space of data using kernel functions may be an infinite-dimensional feature space since data may be well separated in high-dimensional spaces. Kernel principal component analysis (PCA) and kernel linear discriminant analysis (LDA) are two effective methods for achieving effective features of data. To address the outliers of data, $L_1$ norm kernel LDA (Zheng et al., 2014) is developed to achieve the nonlinear discriminant features of data. In unsupervised learning, adopting effective features can contribute to the improvements on the performance of clustering.

The classical k-means method is extended to the kernel k-means method in terms of the kernel trick. To capture multiple features of data, an effective strategy in multiple k-means clustering problems (Yao et al., 2021) is devised to select the optimal kernel from the prespecified kernels, and an alternating minimization method is used to update the coefficients of the kernels and the cluster membership alternatively. Multiple kernel k-means clustering methods with incomplete kernel matrices (MKCIK) (Liu et al., 2020) use imputation and clustering to construct a unified learning framework for the clustering problem. One remarkable characteristic of MKCIK is that a complete base kernel matrix over all the samples is not required.

Exploring the local and relevant information of data points is helpful for achieving discriminant features of data (Flamary et al., 2018; Nie et al., 2023). For each data point, the conditional distribution of the data point can be employed to characterize its local and relevant information. Figure 1 shows that there are three data points in the $\mathbb{X}$ space and nine data points in the $\mathbb{Y}$ space, where each data point in the $\mathbb{X}$ space is relevant to four data points in the $\mathbb{Y}$ space in terms of an appropriate structure such as proximity and topology. In supervised learning, data points with the same color in the $\mathbb{Y}$ space belong to the same class. When the labels of samples are available, in the $\mathbb{Y}$ space, there are four data points whose labels are the same as the label of $x_1$, two data points whose labels are the same as the label of $x_2$, and three data points whose labels are the same as the label of $x_3$. It is clear that the conditional distributions constructed by considering the label information of data points are different from those in unsupervised learning. For data points in the $\mathbb{X}$ space, we can obtain their Dirac measures. Thus, we can explore the unbalanced optimal transport from Dirac measures to conditional measures on two spaces. The optimal

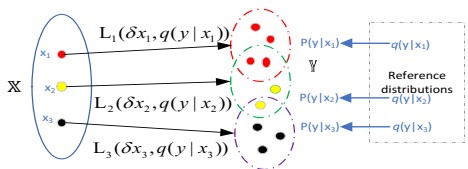

Figure 1: An example of the local and unbalanced optimal transport where the conditional probability is used to characterize the local information of given data points. Each data point in $\mathbb{X}$ can be modeled as a Dirac(point) measure. Data points in the same color in $\mathbb{Y}$ are taken from the same class, and different conditional distributions can be constructed for unsupervised and supervised learning. $L_1(\delta_{x_1}, q(y|x_1))$ denotes the local and unbalanced transport cost from $\delta_{x_1}$ and $q(y|x_1)$ in the original space.

transport (Cuturi, 2013) can be used to describe the relationship between two probability measures, and autoencoders (Berahmand et al., 2024) can explore the latent space of data. Hence, we employ the optimal transport and autoencoders to show how to transport information in an embedding space, which gives a novel framework of learning effective features of data via optimal transport. For each data point, we employ the conditional probability to constrain its transferring range. The merit of using the conditional probability is that varying neighbors of different data points can be explored. To reserve the information of data, we impose the reconstruction error of data on the objective function. In addition, we discuss the properties of our model and extend our model to the clustering problem. Finally, we perform the experiments on a series of data sets. The main contributions of this paper are listed as follows.

- We propose a novel framework to extract the effective and useful features of data in the encoded space. In this framework, we employ conditional distributions to capture the local behaviors of given data points and explore the Kullback-Leibler divergence based on conditional distributions, which can consider prior knowledge of conditional distributions.

- We apply the alternating optimization technique to tackle the proposed model. It is noted that marginal and conditional distributions have closed-form solutions. Moreover, we derive the generalization bound of our model in terms of the Rademacher complexity and generalize our model to find anchors in the encoded space, which is available for the clustering problem.

- We perform a series of experiments on some classification and clustering problems to demonstrate the effectiveness of our model. Moreover, we discuss how to modify our model to make it a classifier that can be used to classify set-valued objects, and this classifier degenerates into the deep nearest-neighbor classifier.

## 2 RELATED WORK

Many feature learning methods based on the deep architectures of neural networks have been developed. Multi-layer learning models (Yuan et al., 2015) have been proposed to deal with the scene recognition problem, and they are available in an unsupervised way. The deep semi-nonnegative matrix factorization (Trigeorgis et al., 2014) can find the latent representation of data in a low-dimensional space, and the new description can improve the clustering performance of data. Deep Fisher discriminant analysis (Diaz Vico & Dorronsoro, 2020) takes advantage of deep neural networks to capture the nonlinear features of data. To deal with sequence data, deep order-preserving Wasserstein discriminant analysis (Su et al., 2022) achieves a nonlinear transformation by maximizing the inter-class distance and minimizing the intra-class distance. The Wasserstein autoencoder (Tolstikhin et al., 2018) was proposed to achieve a generative model of data distributions. However, these feature learning methods do not explore their generalization bounds.

Kernel k-means clustering methods can deal with the nonlinear structure of data in unsupervised learning. For bounded random vectors, the expected excess clustering risk was studied in the work (Maurer & Pontil, 2010). An upper generalization bound of the kernel k-means method in a reduced space (Yin et al., 2022) is derived in terms of the Rademacher complexity. The deep clustering model via the $t$ distribution (DEC) (Xie et al., 2016) has been proposed. The improved DEC (IDEC) (Guo et al., 2017) used autoencoders to enhance the performance of DEC. However, the generalization bounds of DEC and IDEC are not explored.

Optimal transport (OP) and its variants have been applied in many fields such as machine learning and computer vision(Fatras et al., 2021; Montesuma et al., 2025). A unified framework with entropic regularization (Benamou et al., 2015) was proposed to approximate solutions to OT-related linear programs. The regularization facilitates the application of straightforward yet highly effective iterative Kullback-Leibler (KL) projection methods. Unbalanced Gromov-Wasserstein formulations (Sejourne et al., 2021)are constructed to enable the comparison of metric spaces that are endowed with any positive measures. Fused unbalanced Gromov Wasserstein (Thual et al., 2022) is employed for inter-subject alignment. New optimal transport formulation (Manupriya et al., 2024) is derived by utilizing kernelized least-squares terms from joint samples, which ensures that the marginals of the transport plan align with the empirical conditional distributions. The neural network is also used to model the continuous conditional probability.

## 3 LOCAL AND UNBALANCED OPTIMAL TRANSPORT FOR FEATURE LEARNING

### 3.1 PRELIMINARIES

Let two random vectors $X \in R^m$ and $Y \in R^m$ be taken from two probability spaces $(\mathbb{X}, \mu)$ and $(\mathbb{Y}, \nu)$. The $L_r$ norm of a vector $a = (a_1, \cdots, a_m)$ is denoted by $\|a\|_r = \sqrt[r]{\sum_{i=1}^{m} |a_i|^r}$. For measures $\mu$ and $\nu$ corresponding to $X$ and $Y$, the Wasserstein distance with the order $\bar{r}$ is defined as (Courty et al., 2017; Lin & Chan, 2023): $W^{\bar{r}}(\mu, \nu) = \inf_{\pi \in \Pi(\mu, \nu)} \int_{\mathbb{X} \times \mathbb{Y}} \rho(x, y)^{\bar{r}} d\pi(x, y)$ where $\Pi(\mu, \nu)$ is the set of probability measures on $\mathbb{X}$ and $\mathbb{Y}$ with marginal measures $\mu$ and $\nu$, and $\rho(x, y)$ denotes the distance between $x \in R^m$ and $y \in R^m$. $W^{\bar{r}}(\mu, \nu)$ is the potential cost of moving mass from $\mu$ and $\nu$, and the optimal solution provides the optimal transport plan. In real applications, we usually obtain some sampled points in terms of probability measures $\mu$ and $\nu$. That is, $\mu$ and $\nu$ are two discrete measures with a finite number of support points. Thus, $\mu = \sum_{i=1}^{n} a_i \delta_{x_i}$ and $\nu = \sum_{j=1}^{k} b_j \delta_{y_j}$, where $\delta_{x_i}$ denotes the Dirac measure at the point $x_i$, and $a = (a_1, \cdots, a_n)$ and $b = (b_1, \cdots, b_k)$ are vectors in the probability simplex. Assume that $\{x_1, \cdots, x_n\}$ and $\{y_1, \cdots, y_k\}$ are sampled data points from $\mu$ and $\nu$, respectively. To allow some mass variations, unbalanced optimal transport is defined by relaxing marginal constraints in terms of some regularization terms (Chizat et al., 2016), denoted by

$$\inf_p \sum_{i=1}^{n} \sum_{j=1}^{k} \rho(x_i, y_j)^{\bar{r}} p_{ij} + \lambda_1 \mathrm{KL}(pe_k||a) + \lambda_2 \mathrm{KL}(p^T e_n||b), \tag{1}$$

where $p$ is an $n \times k$ matrix, $p_{ij}$ is the nonnegative element of $p$ at row $i$ and column $j$, $\mathrm{KL}(pe_k||a)$ is the Kullback-Leibler (KL) divergence (Bishop, 2007; Zhang et al., 2024) between $pe_k$ and $a$, and $e_k$ is a $k$-dimensional vector whose elements are 1.

## 3.2 PROBLEM FORMULATION

As shown in Figure 1, each data point in a space may locally or semantically correlate with many data points in a space, and conditional distributions can characterize the information of given data points. The theory of optimal transport provides a possible scheme for the movement of data points. Autoencoders facilitate feature formations in an embedding space. For autoencoders, let $f_\theta(x) \in R^d$ be an encoder with parameter $\theta$ and its decoder be $\bar{f}_{\bar{\theta}}(z)$ with parameter $\bar{\theta}$. The functions $g_\phi(y)$ and $\bar{g}_{\bar{\phi}}(z)$ consist of another autoencoder. *Even if the original spaces have different dimensions, we employ the encoders to achieve features with the same dimensions, which makes the optimal transport feasible in the encoded spaces*. In encoded spaces, we obtain the Dirac measure at $f_\theta(x_i)$, denoted by $\delta_{f_\theta(x_i)}$. We employ the conditional distribution $p(g_\phi(y)|x_i)$ to characterize the information depending on $x_i$.

To facilitate the learning of the conditional distribution, we adopt a convex combination of Dirac measures to construct $p(g_\phi(y)|x_i)$. That is, $y$ takes $k$ values and $p(g_\phi(y)|x_i) = \sum_{j=1}^k p_{j|i}\delta_{g_\phi(y_{j|i})}$, where $p_{j|i}$ is a nonnegative coefficient that satisfies $\sum_{j=1}^k p_{j|i} = 1$. $y_{j|i}$ may be semantically relevant to $x_i$. This also ensures that $p(g_\phi(y)|x_i)$ belongs to the Wasserstein space (Victor M. Panaretos, 2020). Thus, the $rth$-order Wasserstein distance between $\delta_{f_\theta(x_i)}$ and $p(g_\phi(y)|x_i)$ with $k$ sampling points can be obtained, denoted by $\bar{W}_i^{\bar{r}} = \min_{p_{j|i}} \sum_{j=1}^k \rho(f_\theta(x_i), g_\phi(y_{j|i}))^{\bar{r}} p_{j|i}$. To effectively learn $p_{j|i}$ in the conditional distributions, we use a special case of the unbalanced optimal transport where $\delta_{f_\theta(x_i)}$ and $p(g_\phi(y)|x_i)$ are explored.

$$\min_{p_{j|i}} L_i := \sum_{j=1}^k \rho(f_\theta(x_i), g_\phi(y_{j|i}))^{\bar{r}} p_{j|i} + \lambda_1 \text{KL}(p_{\cdot|i}||q_{\cdot|i}), \tag{2}$$

where $\text{KL}(p_{\cdot|i}||q_{\cdot|i}) = \sum_{j=1}^k p_{j|i} \ln \frac{p_{j|i}}{q_{j|i}}$, $\sum_{j=1}^k q_{j|i} = 1$, $q_{j|i}$ is the prior probability of transferring $x_i$ to $y_{j|i}$ in the original space, and $y_{j|i}$ is the $jth$ data point that is semantically related to $x_i$. The KL divergence is employed to measure the difference between $\{p_{j|i}\}$ and $\{q_{j|i}\}$. Since we introduce the conditional probability of $x_i$, we can control the transferring range of $x_i$. If $x_i$ is not allowed to be moved to $y_{j|i}$, then we set $q_{j|i} = 0$. The nonnegative hyperparameter $\lambda_1$ controls the tradeoff between the transport cost and the KL divergence. The variables $p_{j|i}(j = 1, \cdots, k)$ need to be optimized, and they implicitly depend on the embedding spaces. $q_{j|i}$ is the prior conditional probability independent of encoded spaces.

Interestingly, optimizing (2) can also be regarded as a proximal algorithm to achieve the proximal operator (Li et al., 2023; Gu et al., 2024; Zhang et al., 2024). Unlike those proximal operators, we explore the conditional distribution in a discrete form. In fact, one may replace the KL term in (2) with the f-divergence (Gonzlez-Snchez, 2022; Gu et al., 2024) between two distributions, which increases the flexibility of the model(see Appendix A.2 ). For computational convenience, we apply the KL divergence in (2). To explore the transport cost of all data points, we define the following model:

$$\min_{(\theta, \phi, p_{j|i}, p_i)} \hat{L}_w := \sum_{i=1}^n L_i p_i + \lambda_2 \text{KL}(p||q), \tag{3}$$

where $p_i = p(x_i)$, $\sum_{i=1}^n p_i = 1$, $\text{KL}(p||q) = \sum_{i=1}^n p_i \ln \frac{p_i}{q_i}$, $q_i$ is the prior probability of $x_i$ independent of embedding spaces, and $\lambda_2$ is a nonnegative hyperparameter.

The first term in (3) denotes the unbalanced transport cost of all data points, and the second term is the KL divergence between $\{p_i\}$ and $\{q_i\}$. If $\{x_i\}$ are sampled from the uniform distribution, i.e., $p_i = 1/n$, we let $\lambda_2 = 0$ since the KL term is constant. The introduction of $q_{j|i}$ and $q_i$ can help us explore prior knowledge of data from the original space. If no prior knowledge of data is available, $q_{j|i}$ and $q_i$ may take the uniform distribution. Here, we take the long-tailed student $t$ distribution as the probability of moving from $x_i$ to $y_{j|i}$ (Xie et al., 2016) in the original space, denoted by $q_{j|i} = \frac{(1+\rho(x_i,y_{j|i})^{\bar{r}})^{-1}}{\sum_{j=1}^k (1+\rho(x_i,y_{j|i})^{\bar{r}})^{-1}}$. Note that $q_{j|i}$ depends on the original features instead of the encoded features. Generally, $q_{j|i}$ reflects information in the original space, but $p_{j|i}$ can be learned from the encoded space.

Unlike the optimal transport theory, we decompose the joint distribution into the product of two distributions, i.e., $p_{i,j} = p_i p_{j|i}$. Moreover, we utilize the conditional KL divergence as the regularization term by introducing prior conditional probabilities of data points. Note that trivial solutions

of $\theta$ and $\phi$ may be obtained if we do not impose additional constraints on encoders. In order to address this problem, we add the reconstruction error of data to the objective function of (3) by using decoders. Thus, we define the following model:

$$\min_{(\theta,\bar{\theta},\phi,\bar{\phi},p_{j|i},p_i)} \hat{L} := \hat{L}_w + \lambda_3 \sum_{i=1}^n \|x_i - \bar{f}_{\bar{\theta}} f_\theta(x_i)\|_2 p_i$$

$$+ \lambda_4 \sum_{i,j=1}^{n,k} p_i p_{j|i} \|y_{j|i} - \bar{g}_{\bar{\phi}} g_\phi(y_{j|i})\|_2, \tag{4}$$

where $\lambda_i (i = 3, 4)$ are nonnegative hyperparameters. The last two terms in (4) involve the reconstruction errors of $x_i$ and $y_{j|i}$. The continuous version of (4) can be found in Appendix A.3. From (4), we find that the loss function in the proposed model consists of the transport cost, reconstruction errors of data and additional regularization terms. The framework is generic since we do not give specific autoencoders and any transport cost can be used to replace $\rho(\cdot)$. Note that in the above model, we assume that $\{x_i\}$ and $\{y_{j|i}\}$ adopt different encoders and decoders. In fact, when $\{x_i\}$ and $\{y_{j|i}\}$ are sampled from the same data source, we can take the same encoders and decoders. **In this paper, we only consider that $\{x_i\}$ and $\{y_{j|i}\}$ take the same encoders and decoders, but we reserve more general notations for future extensions of our framework for different dimensions of features from two data sources.** Since we consider the conditional distribution of $x_i$, we use it to describe the local information of $x_i$. That is, $y_{j|i}(j = 1, \cdots, k)$ are taken from the $k$ neighbors of $x_i$. In supervised learning, we allow $y_{j|i}$ to be taken from the samples whose labels are the same as the label of $x_i$. If $\{x_i\}$ and $\{y_{j|i}\}$ are taken from the same data source, we let $f_\theta = g_\phi$, $\bar{f}_{\bar{\theta}} = \bar{g}_{\bar{\phi}}$, and $\lambda_4 = 0$.

### 3.3 OPTIMIZATION

Note that there are several groups of parameters to be optimized in our model. Moreover, some parameters such as the conditional probability have additional constraints. Thus, the model of (4) is a constrained and non-convex optimization problem. To solve our model, we resort to the alternating optimization technique. Specifically, we alternatively optimize a group of variables by fixing other groups of optimization variables. In the following, we will demonstrate how to divide these variables into several groups and how to optimize them.

(a): Update $p_{j|i}$ by fixing other variables. In this step, when we fix $\theta, \bar{\theta}, \phi, \bar{\phi}$, and $p_i$, we solve the following model:

$$\min_{p_{j|i}} \sum_{i,j=1}^{n,k} \rho(f_\theta(x_i), g_\phi(y_{j|i}))^{\bar{r}} p_i p_{j|i} + \lambda_1 \sum_{i=1}^n p_i \text{KL}(p_{\cdot|i} \| q_{\cdot|i}) +$$

$$\lambda_4 \sum_{i,j=1}^{n,k} p_i p_{j|i} \|y_{j|i} - \bar{g}_{\bar{\phi}} g_\phi(y_{j|i})\|_2. \tag{5}$$

It is noted that (5) is a convex optimization problem. It is of interest to note that it has a closed-form solution, denoted by $p_{j|i} = \frac{q_{j|i} exp(-(L_{j|i}^{op}+L_{j|i}^{re})/\lambda_1)}{\sum_{j=1}^k q_{j|i} exp(-(L_{j|i}^{op}+L_{j|i}^{re})/\lambda_1)}$, where $L_{j|i}^{op} = \rho(f_\theta(x_i), g_\phi(y_{j|i}))^{\bar{r}}$ and $L_{j|i}^{re} = \lambda_4 \|y_{j|i} - \bar{g}_{\bar{\phi}} g_\phi(y_{j|i})\|_2$. The conditional probability $p_{j|i}$ can be obtained by using the lemma in Appendix A.4 ).

(b): Update $p_i$ by fixing other variables. Given $\theta, \bar{\theta}, \phi, \bar{\phi}$, and $p_{j|i}$, we achieve $p_i$ by solving the following problem:

$$\min_{p_i} \sum_{i,j=1}^{n,k} \rho(f_\theta(x_i), g_\phi(y_{j|i}))^{\bar{r}} p_i p_{j|i} + \lambda_1 \sum_{i=1}^n p_i \text{KL}(p_{\cdot|i} \| q_{\cdot|i}) + \lambda_2 \text{KL}(p \| q) +$$

$$\lambda_3 \sum_{i=1}^n \|x_i - \bar{f}_{\bar{\theta}} f_\theta(x_i)\|_2 p_i + \lambda_4 \sum_{i,j=1}^{n,k} p_i p_{j|i} \|y_{j|i} - \bar{g}_{\bar{\phi}} g_\phi(y_{j|i})\|_2. \tag{6}$$

It is observed that the objective function in (6) is convex. The closed-form solution is denoted by $p_i = \frac{q_i exp(-(L_i^{op}+L_i^{enre})/\lambda_2)}{\sum_{i=1}^n q_i exp(-(L_i^{op}+L_i^{enre})/\lambda_2)}$, where $L_i^{op} = \sum_{j=1}^k \rho(f_\theta(x_i), g_\phi(y_{j|i}))^{\bar{r}} p_{j|i}$ and $L_i^{enre} = \lambda_4 \sum_{j=1}^k p_{j|i} \|y_{j|i} - \bar{g}_{\bar{\phi}} g_\phi(y_{j|i})\|_2 + \lambda_3 \|x_i - \bar{f}_{\bar{\theta}} f_\theta(x_i)\|_2 + \text{KL}(p_{\cdot|i} \| q_{\cdot|i})$.

(c): Update $\theta, \bar{\theta}, \phi, \bar{\phi}$ by fixing other variables. In this step, we try to learn the parameters of autoencoders. Specifically, we solve the following optimization problem:

$$
\min_{(\theta, \bar{\theta}, \phi, \bar{\phi})} \sum\nolimits_{i,j=1}^{n,k} \rho(f_\theta(x_i), g_\phi(y_j))^{\bar{r}} p_i p_{j|i} + \lambda_3 \sum\nolimits_{i=1}^{n} p_i \|x_i - \bar{f}_{\bar{\theta}} f_\theta(x_i)\|_2 +
$$

$$
\lambda_4 \sum\nolimits_{i,j=1}^{n,k} p_i p_{j|i} \|y_{j|i} - \bar{g}_{\bar{\phi}} g_\phi(y_{j|i})\|_2. \tag{7}
$$

Note that the objective function in (7) is nonconvex. We cannot obtain the global optimal solution. We generally update these parameters of models through the chain rule in the framework of neural networks. In this work, we resort to automatic differentiation to learn these parameters.

For completeness, we summarize the main steps of solving the proposed model in Algorithm 1. It is found that solving (7) involves the computational complexity of $O(H_1^2 H_2 n)$ in each iteration, solving (5) involves $O(nk(m + d))$ and solving (6) gives $O(n(m + d))$, where $H_1$ is the maximum number of hidden units of layers and $H_2$ is the number of layers.

---

**Algorithm 1:** Optimization algorithm to (4)

1: Given $\lambda_i$, $q_{j|i}$, $q_i$, and initialize $p_i = q_i$, $p_{j|i} = q_{j|i}$
2: **For** t=1 to T **do**
   2.1: solve (7) to achieve its parameters $(\theta, \bar{\theta}, \phi, \bar{\phi})$;
   2.2: solve (5) to achieve $p_{j|i}$;
   2.3: solve (6) to achieve $p_i$;
3: Output: the encoders and decoders.

---

### 3.4 THEORETICAL ANALYSIS OF OUR MODEL

In this subsection, we theoretically analyze some properties of our model. There are several parameters in our models. We observe that $\lim_{\lambda_1 \to \infty} p_{j|i} = q_{j|i}$ and $\lim_{\lambda_2 \to \infty} p_i = q_i$ if $p_{j|i}$ and $p_i$ are defined in subsection 3.3. This indicates that if parameters $\lambda_1$ and $\lambda_2$ approach the positive infinity, $p_{j|i}$ and $p_i$ will have the same distributions as prior distributions. If prior distributions are uniform distributions, the optimal transport plan will be uniform distributions. In such a case, the objective function of our model makes the trade-off between the reconstruction error and the transport cost. Note that when deriving the generalization bound of our model, we do not consider the expectation with respect to the random variable $Y$. Here we assume that $Y$ has the support consisting of $k$ data points. For given $x_i$, we need to find $k$ data points $y_{1|i}, \cdots, y_{k|i}$. These $k$ data points are varying for different $x_i$. Evidently, it is different from the fixed sampled points $y_1, \cdots, y_k$ in the optimal transport theory. To explore the effect of the parameters of networks, we study the generalization bound of our model based on the assumption that $S = \{x_1, \cdots, x_n\}$ are independent and identically distributed samples, i.e., $p_i = \frac{1}{n}$. First we define the empirical loss as done in (Maurer & Pontil, 2010) when $\lambda_4$ takes the zero value.

$$
\hat{L}_S(\theta, \bar{\theta}) := \min_{p_{j|i}} \frac{1}{n} \{ \sum\nolimits_{i,j=1}^{n,k} p_{j|i} \rho(f_\theta(x_i), f_\theta(y_{j|i}))^{\bar{r}} +
$$

$$
\lambda_1 \sum\nolimits_{i=1}^{n} \mathrm{KL}(p_{\cdot|i} \| q_{\cdot|i}) + \sum\nolimits_{i=1}^{n} \lambda_3 \|x_i - \bar{f}_{\bar{\theta}} f_\theta(x_i)\|_2 \}. \tag{8}
$$

Note that there are several differences between Equations (4) and (8). Here, we let $f_\theta = g_\phi$ and $\bar{f}_{\bar{\theta}} = \bar{g}_{\bar{\phi}}$. Moreover, we do not consider the reconstruction error of $y_{1|i}, \cdots, y_{k|i}$ since they are taken from the space that $x_i$ belongs to. In addition, we assume that $p_i$ in (8) is taken from the uniform distribution. Let $L(\theta, \bar{\theta})$ be the expected loss corresponding to (8). We make the assumptions. *A0: the distance measure has the form of $\rho(x, y) = \varphi(x - y)$ and $\varphi(x)$ has the Lipschiz constant $\ell$; A1: $x_i$ and $y_{j|i}$ are bounded, i.e., $\exists M$ such that $\|x_i\|_2 \le M$ and $\|y_{j|i}\|_2 \le M$; A2: $\|\bar{f}_{\bar{\theta}}\|_2 \le M$ and $\|f_\theta\|_2 \le M$ hold for parameters $\bar{\theta}$ and $\theta$ in a parameter space; A3: if $q_{j|i} = 0$, $p_{j|i} = 0$.*

The assumption A0 holds if the metric is induced by the norm in a normed space and the data are taken from a compact space. For example, $\rho(x, y)$ takes the form of the $L_r$ norm. The assumptions A1 and A2 are reasonable since the data we deal with are bounded. The assumption A3 ensures that the KL divergence is well defined. Now we show the uniform deviation bound of the objective function in (8) by using the following theorem.

**Theorem 1**. *Under the above assumptions, with probability at least $1 - \tau$, the following inequality holds for $\theta$ and $\bar{\theta}$ in proper parameter spaces:*

$$\hat{L}_S(\theta, \bar{\theta}) \leq L(\theta, \bar{\theta}) + 4\sqrt{2}M_1R_1 + 2\sqrt{2}R_2 + \chi_1\sqrt{\frac{-log\tau}{2n}} \tag{9}$$

*where $M_1 = \bar{r}(2M\ell)^{\bar{r}-1}\ell$, $\quad \chi_1 = \frac{2(2M)^{\bar{r}}+4\lambda_3 M}{n}$, $R_1 = E_{S,\sigma}\frac{1}{n}\sup_\theta \sum_{t=1}^d |\sum_{i=1}^n \sigma_{it}(f_\theta(x_i))_t|$, $R_2 = E_{S,\sigma}\frac{1}{n}\sup_{\theta,\bar{\theta}}\lambda_3 \sum_{t=1}^m |\sum_{i=1}^n \sigma_{it}(\bar{f}_{\bar{\theta}}f_\theta(x_i))_t|$, $\sigma_{it}$ denotes the Rademacher random variable, $E_S$ denotes the expectation with respect to $S$, and $f_\theta(x_i)_t$ denotes the t-th element of the vector $f_\theta(x_i)$.*

The proof of Theorem 1 can be found in Appendix A.5. In (9), $R_1$ denotes the Rademacher complexity of the encoder $f_\theta(\cdot)$, and $R_2$ denotes the Rademacher complexity of the encoder-decoder $\bar{f}_{\bar{\theta}}f_\theta(\cdot)$. In the case of a single-layer linear network, if the parameters of the network satisfy $\theta^T\theta = I_d$ and $\bar{\theta} = \theta^T$, then we have $R_1 \leq kdM/\sqrt{n}$ and $R_2 \leq dM/\sqrt{n}$. It has been proved in (Truong, 2019) that the Rademacher complexity of deep learning models is of order $O(1/\sqrt{n})$ under proper conditions. Thus, $R_1$ and $R_2$ have the order of $O(1/\sqrt{n})$.

## 3.5 EXTENSIONS TO THE CLUSTERING PROBLEM

In the above section, we assume that $x_i$ is transported to data points $y_{1|i}, \cdots, y_{k|i}$. These data points are taken from the class of $x_i$ or from $k$-neighbors of $x_i$. This implicitly uses prior knowledge from the original data. **Without using prior knowledge, can they be learned from data via some optimization methods**? This may be a trivial thing since the number of $\{y_{j|i}|i = 1, \cdots, n, j = 1, \cdots, k\}$ is much bigger than that of $\{x_i|i = 1, \cdots, n\}$ as shown in Figure 1. To avoid triviality, we can impose additional constraints on $\{y_{j|i}\}$ to reduce the number of $\{y_{j|i}\}$. In supervised learning, we may consider that the data points in the same class are transported to unknown data points(anchors). That is, $y_{j|s} = y_{j|t}$ if $x_s$ and $x_t$ are from the same class. In unsupervised learning where the labels of samples are not available, we may consider the case where all the data points $\{x_i|i = 1, \cdots, n\}$ are transported to unknown data points $\{y_j|j = 1, \cdots, k\}$, i.e., $y_j = y_{j|1} = \cdots = y_{j|n}$. Thus, the conditional distribution is denoted by $p(g_\phi(y)|x_i) = \sum_{j=1}^k p_{j|i}\delta_{g_\phi(y_j)}$, where $p_{j|i}$ and $y_j$ need to be learned. Instead of finding $\{y_j|j = 1, \cdots, k\}$ in the original space, we explore unknown data points in an embedding space and let $z_j = g_\phi(y_j)(j = 1, \cdots, k)$. Since we directly look for $\{z_j\}$ in the encoded space, we do not need to consider the encoder $g_\phi$ and the decoder $\bar{g}_{\bar{\phi}}$. Thus, the following model is formulated to learn $\{z_j\}$ in an embedding space of data in an unsupervised way.

$$\min_{(\theta, \bar{\theta}, p_{j|i}, p_i, z_j)} \hat{L} := \sum_{i,j=1}^{n,k} p_i \rho(f_\theta(x_i), z_j)^{\bar{r}} p_{j|i} + \lambda_1 \sum_{i=1}^n p_i \mathrm{KL}(p_{\cdot|i}||q_{\cdot|i})+$$
$$\sum_{i=1}^n \lambda_3 p_i \|x_i - \bar{f}_{\bar{\theta}}f_\theta(x_i)\|_2 + \lambda_2 \mathrm{KL}(p||q). \tag{10}$$

Note that $z_1, \cdots, z_k$ are optimization variables in an embedding space. We refer to $z_1, \cdots, z_k$ as anchors. These anchors can also be taken as the cluster centroids of data in the embedding space if $k$ is equal to the number of clusters. In such a case, the conditional probability $p_{j|i}$ can be regarded as the probability of $x_i$ closing to $z_j$. Compared with some embedding clustering methods(Xie et al., 2016; Guo et al., 2017), our model explores the weights of samples. We also employ the alternating optimization method to solve (10), which can be found in Appendix A.6. The conditional probability $p_{j|i}$ and marginal probability $p_i$ have closed-form solutions in each step. In such a case, we can learn the anchors (centroids) in the embedding space by using autoencoders. The main aim of designing our model of (10) is to obtain features in the embedding space in an unsupervised way. If $k = n$, we can train our model to obtain the embedding points of $n$ data points in terms of prior knowledge $p_i$ and $q_{j|i}$. Then we can employ encoders to obtain the embedding of any data point. Here, we employ (10) to learn the embedding space of data and perform the possible clustering tasks in the embedding space. In fact, pretrained autoencoders may be employed to initialize the weights of autoencoders. When data points are independent and identically distributed, we can explore the

generalization bound of our model of (10). To this end, we define the following empirical loss.

$$\hat{L}_S^c(\theta, \bar{\theta}, z_j) := \min_{p_{j|i}} \sum_{i,j=1}^{n,k} \frac{1}{n} \rho(f_\theta(x_i), z_j)^{\bar{r}} p_{j|i} + \frac{\lambda_1}{n} \sum_{i=1}^{n} \mathrm{KL}(p_{\cdot|i}||q_{\cdot|i}) + \sum_{i=1}^{n} \frac{\lambda_3}{n} \|x_i - \bar{f}_{\bar{\theta}} f_\theta(x_i)\|_2.$$

$$(11)$$

Let $L^c(\theta, \bar{\theta}, z_j)$ be the expected loss corresponding to $\hat{L}_S^c(\theta, \bar{\theta}, z_j)$. We give the following theorem to characterize the generalization bound of (11).

**Theorem 2**. *As with the assumptions in (10), with probability at least $1 - \tau$, the following inequality holds for $\theta, \bar{\theta}, z_j$ in proper parameter spaces:*

$$\hat{L}_S^c(\theta, \bar{\theta}, z_j) \leq L^c(\theta, \bar{\theta}, z_j) + 2\sqrt{2}M_1 R_1 + 2\sqrt{2}R_2 + \frac{\chi_1 + \chi_2}{\sqrt{n}} \tag{12}$$

*where $M_1 = \bar{r}(2M\ell)^{\bar{r}-1}\ell$, $\chi_1 = \frac{2(2M)^{\bar{r}} + 4\lambda_3 M}{n}\sqrt{\frac{-\ln \tau}{2}}$, $R_1 = E_{S,\sigma} \frac{1}{n} \sup_\theta \sum_{t=1}^{d} |\sum_{i=1}^{n} \sigma_{it}(f_\theta(x_i))_t|$,*

*$R_2 = E_{S,\sigma} \frac{1}{n} \sup_{\theta,\bar{\theta}} \lambda_3 \sum_{t=1}^{m} |\sum_{i=1}^{n} \sigma_{it}(\bar{f}_{\bar{\theta}} f_\theta(x_i))_t|$, $\chi_2 = 2\sqrt{2}M_1 M dk$, $\sigma_{it}$ denotes the Rademacher random variable, $E_S$ denotes the expectation with respect to $S$, and $f_\theta(x_i)_t$ denotes the $t$-th element of the vector $f_\theta(x_i)$. The proof of Theorem 2 can be found in Appendix A.7.* **Compared to (10), an additional term $\chi_2$ appears in (12) due to the optimization of anchors (centroids).** It is found that the upper bound of the empirical loss depends on the number of anchors. Our generalization bound has the same convergence speed as the k-means algorithm (Maurer & Pontil, 2010).

## 4 EXPERIMENTAL RESULTS

### 4.1 EXPERIMENTS ON SUPERVISED LEARNING

We perform the experiments on some data sets to obtain effective representations of features for classification tasks. In experiments, $x_1, \cdots, x_n$ consist of the training set and $y_{j|i}(j = 1, \cdots, k)$ are taken from the samples that have the same label as $x_i$. It is found that some face data sets belong to the small-sample-size problem since the number of each class in the training set is much smaller than the dimension of the samples. When our model adopts one linear layer, we refer to our model as local and unbalanced optimal transport (LUOP-L). When our model contains several linear layers and ReLU functions, we refer to our model as the local and unbalanced optimal transport (LUOP). The dimension of encoded spaces in our model is equal to the number of classes. We compare our model with several kernel-based methods including kernel discriminant analysis (K-DA) (Zheng et al., 2014), kernel discriminant analysis based on the $L_1$ norm (KDAL1) (Zheng et al., 2014) and regularized kernel discriminant analysis (RKDA) (Diaz Vico & Dorronsoro, 2020). In addition, autoencoders(AE), autoecoders with Laplacian matrix(AEL) (Yang et al., 2017), deep Fisher discriminant analysis (DFDA) (Diaz Vico & Dorronsoro, 2020) and deep Wasserstein discriminant analysis (DWDA) (Su et al., 2022) are tested. Since our model is employed to explore the latent space in supervised learning, we adopt the nearest neighbor (NN) classifier with the Euclidean norm. Experimental results on the data sets are shown in Table 1 and experimental details are in Appendix A.8.

Table 1: Error rates $(\%)$ of various methods and their standard deviations on data sets

| data sets | KDA | KDAL1 | RKFDA | DFDA | DWDA | AE | AEL | LUOP | LUOP-L |
|---|---|---|---|---|---|---|---|---|---|
| Dna | 10.18±2.37 | 9.74±2.46 | 9.56±2.35 | 9.41±3.05 | 9.49±2.27 | 9.62±1.17 | 9.55±2.06 | 9.58±2.47 | **9.21±2.35** |
| Pendigits | 7.36±1.29 | 6.25±1.04 | 6.17±3.24 | 6.27±2.08 | 6.32±2.38 | 6.34±1.98 | 6.21±1.04 | 6.20±1.45 | **6.05±1.38** |
| Iris | 4.00±2.28 | 3.33±2.04 | 3.33±2.04 | 4.00±2.28 | 3.33±2.04 | 4.25±1.08 | 3.67±1.48 | 3.33±2.04 | **2.56±1.78** |
| Satimage | 24.57±2.26 | 24.38±2.67 | 24.69±3.05 | 16.77±2.59 | 16.86±2.62 | 18.51±3.25 | 17.21±2.05 | 23.46±2.77 | **16.57±2.61** |
| Waveform | 22.26±1.72 | 20.34±1.51 | 20.11±1.47 | 20.19±1.52 | 19.87±2.24 | 21.24±1.67 | 19.96±1.82 | 20.21±1.85 | **19.02±1.76** |
| ORL | 8.76±2.12 | 8.53±2.09 | 8.36±2.24 | 10.46±2.37 | 10.55±2.16 | 10.57±2.70 | 10.37±2.15 | **8.21±2.45** | 10.38±1.92 |
| Yale | 7.52±3.50 | 7.44±3.95 | 7.26±3.41 | 11.47±3.09 | 11.90±3.51 | 11.93±3.41 | 11.58±3.57 | **7.20±1.05** | 11.93±1.55 |
| UMIST | 8.97±2.25 | 8.76±2.34 | 8.45±3.02 | 10.56±3.50 | 10.78±3.05 | 10.90±4.001 | 10.72±4.21 | **8.21±2.92** | 10.33±2.06 |
| COIL | 8.45±2.21 | 9.43±1.65 | 8.22±1.69 | 8.13±2.02 | 8.06±1.72 | 8.29±1.62 | **8.04±1.47** | 8.19±1.98 | 8.08±1.23 |
| MSRA | 10.12±1.05 | 9.56±0.98 | 9.35±0.97 | 9.43±1.02 | 9.46±1.15 | 9.56±0.57 | 9.43±0.81 | **9.21±1.98** | 9.72±1.09 |

From Table 1, we can see that deep learning models such as DFDA, DWDA and LUOP-L perform poorly on ORL, Yale and UMIST data sets. This comes from the fact that overfitting occurs since there are not enough training samples to learn the parameters of deep learning models. However,

LUOP-L obtains better performance than other methods on these face data sets. It is found that KDAL1 is superior to KDA on these data sets since KDAL1 is robust to outliers. DFDA and DWDA do not explore the reconstruction error of samples, whereas LUOP makes use of the reconstruction error of the samples. Overall, it is reasonable to use conditional distributions to locally transport data points in an encoded space.

## 4.2 CLUSTERING EXPERIMENTS

We verify the proposed model on some data sets in terms of clustering tasks. We use the normalized mutual information (NMI) to show the performance of the clustering methods. We also implement kernel k-means (KKM)(Paul et al., 2022), kernel fuzzy k-means (KFKM)(Paul et al., 2022), kernel power k-means(KPKM) (Paul et al., 2022), the deep clustering model based on the $t$ distribution (DEC) (Xie et al., 2016), the improved DEC(IDEC) based on autoencoders (Guo et al., 2017), and the deep fuzzy $k$-means method (DFKM) (Zhang et al., 2020). Since the aim of our framework of (10) is to search for the embedding space of data in terms of autoencoders, we can use any clustering method after the embedding space of data is obtained. Here, we perform the spectral clustering on obtained features, where the number of neighbors is 5. In such a case, we refer to our model as the local and unbalanced optimal transport plus spectral clustering (LUOPSC). Table 2 shows the NMI of various methods where we list the best result of each method. From Table 2, we note that our model is superior to other models since we optimize anchors to learn features in an embedding space. Note that deep-learning models such as DEC, IDEC and DFKM jointly learn data embedding and clustering. KKM, KFKM, and KPKM make use of kernel functions to learn the embedding space. It is found that the features based on deep learning models are better than those from kernel functions. The experimental results show that feature learning via optimal transport and autoencoders is effective in unsupervised learning.

Table 2: NMI values (%) of various methods and their standard deviations on data sets

| data sets | KKM | KFKM | KPKM | DEC | IDEC | DFKM | AE | AEL | LUOPSC |
|---|---|---|---|---|---|---|---|---|---|
| Dna | 40.25±2.46 | 42.55±3.17 | 46.34±2.81 | 49.22±2.70 | 49.46±3.10 | 48.26±2.92 | 48.47±2.76 | 49.58±2.31 | **50.21±2.86** |
| Pendigits | 72.46±2.23 | 73.82±1.90 | 73.25±2.24 | 72.35±2.72 | 72.36±2.67 | 77.85±1.90 | 71.12±1.89 | 78.26±3.04 | **80.55±1.79** |
| Iris | 76.74±2.50 | 77.83±2.89 | 78.35±3.01 | 80.79±2.68 | 81.44±2.71 | 80.25±3.05 | 80.25±3.44 | 85.38±2.76 | **88.46±2.96** |
| Satimage | 62.33 ±3.05 | 62.25±3.23 | 63.35±3.19 | 65.56±3.52 | 65.26±3.43 | 68.33±3.66 | 64.32±3.01 | 69.27±4.05 | **70.05±3.25** |
| Waveform | 27.22±2.04 | 28.46±2.17 | 30.51±2.51 | 30.11±2.62 | 34.61±2.73 | 35.14±2.29 | 32.05±2.02 | 36.25±2.09 | **38.16±2.14** |
| ORL | 62.45±2.78 | 63.64±3.07 | 65.57±3.12 | 70.65±2.46 | 70.24±2.71 | 69.33±2.66 | 69.37±2.08 | 75.65±3.41 | **80.63±3.41** |
| Yale | 60.22 ±4.52 | 61.25±4.23 | 63.33±4.62 | 65.12±4.02 | 64.18±4.19 | 65.26±4.31 | 64.49±3.03 | 70.26±4.89 | **73.22±4.51** |
| UMIST | 72.35±3.12 | 72.33±3.25 | 76.59±3.28 | 81.26±3.42 | 80.36±3.30 | 82.35±3.66 | 8.90±3.75 | 84.62±3.2 | **87.18±3.20** |
| COIL | 78.69±2.21 | 80.42±2.32 | 82.62±2.44 | 90.12±2.50 | 90.25±2.36 | 86.26±2.68 | 89.52±3.06 | 91.26±2.09 | **92.35±2.23** |
| MSRA | 56.12±2.02 | 58.24±2.12 | 57.22±2.29 | 60.22±2.30 | 62.21±2.19 | 59.23±2.26 | 60.17±2.56 | **62.30±2.53** | 61.26±2.25 |

## 4.3 EXPERIMENTS ON TWO LARGE-SCALE DATA SETS

We find that on small-scale data sets, using a one-layer network sometimes obtains much better performance than using multiple-layer networks. Does this phenomenon occur on large-scale data sets? In the following experiments, we find that this phenomenon does not occur. Here we select two large-scale image data sets (MNIST and FashionMNIST) to evaluate the proposed model. The aim of using these two data sets is that we do not need to employ complex neural networks to achieve relatively good performance. Unlike the deep learning models based on data augmentation, we only use our autoencoders to achieve the embedding features. The training samples are employed to select the parameters of models and test samples are used to measure the performance of models. In our experiments, we adopt a large batch size of 2000. Since there are a large number of samples in the training set, we employ the class-mean classifier in the classification task. In the kernel-based methods, 100 anchors taken from the $k$-means algorithms are employed to compute kernel matrices since computing kernel matrices for all the samples is impossible. Table 3 lists the experimental results from classification and clustering tasks. From Table 3, we note that the performance of LUOP in the classification experiments is much better than that of LUOP-L. It shows that using multiple-layer networks is beneficial for large-scale data sets. It is clear that our method is superior to other methods since we explore the transferring range of data in the embedding space via the unbalanced optimal transport. In the clustering experiments, we observe that our model outperforms other models since we employ optimal transport and autoencoders to learn the anchors.

Table 3: Classification (error rates) and clustering (NMI) on two large-scale data sets

| classification | KDA | KDAL1 | RKFDA | DFDA | DWDA | AE | AEL | LUOP-L | LUOP |
|---|---|---|---|---|---|---|---|---|---|
| MNIST | 10.30±2.12 | 12.15±2.57 | 9.55±1.86 | 9.39±2.32 | 8.86±2.73 | 9.06±2.13 | 8.47±2.36 | 9.19±2.26 | **8.21±2.31** |
| Fashion | 12.30±3.49 | 14.36±3.89 | 11.79±4.01 | 11.22±3.37 | 10.35±3.58 | 12.03±3.41 | 9.92±3.40 | 10.41±3.3 | **9.21±3.17** |
| Clustering | KKM | KFKM | KPKM | DEC | IDEC | DFKM | AE | LAE | LUOPSC |
| MNIST | 54.33±2.53 | 59.40±2.67 | 58.37±2.49 | 67.46±2.56 | 79.21±2.73 | 70.23±2.12 | 68.47±2.40 | 75.62±2.26 | **81.24±2.51** |
| Fashion | 46.62±3.72 | 47.39±3.69 | 50.28±3.10 | 54.35±3.53 | 56.45±3.44 | 54.37±2.76 | 54.46±3.08 | 60.02±2.97 | **62.08±3.16** |

## 4.4 CLASSIFICATION OF SET-VALUED OBJECTS

Here we modify our model to make it capable of handling set-valued classification problems. For set-valued classification problems, each object contains many examples. Unlike previous experiments, we assume that the set $\{x_1, \cdots, x_n\}$ is a set-valued object containing $n$ examples in the validation set or test set. For the data point $x_i$, we can obtain its $k$ neighbours $y_{1|i}, \cdots, y_{k|i}$ and these $k$ neighbours are from the training set. Since we know the labels of $y_{1|i}, \cdots, y_{k|i}$ in the training set, we assign the label of $x_i$ to the label of $y_{j|i}$ with the largest $p_{j|i}(j = 1, \cdots, k)$. Thus, we obtain the label of each example in a set-valued object. Finally, the majority voting strategy is employed to achieve the label of the set-valued object. We refer to our model as the local and the unbalanced optimal transport classifier (LUOPC). Our model will degenerate into the deep nearest-neighbor classifier if each object only contains an example and the parameter $\lambda_1$ approaches the positive infinity. Here we need to use the validation set to learn the embedding space of data and hyperparameters. In the test stage, we fix the parameters of autoencoders and optimize $p_{j|i}$. We test LUOPC on two medical image sets in binary classification problems (Yang et al., 2021). We use 780 images from the breast image set and 4708 images from the pneumonia image set. To evaluate the performance of LUOPC, we compare it with several set-valued data classification methods such as the second-order cone programming (SOCP) approach (Shivaswamy et al., 2006), the sparse approximated nearest point (SANP) method (Hu et al., 2011), regularized collaborative representation classification (RCRC) (Zhu et al., 2014), support measure machines (SMMs) (Muandet et al., 2012), and support function machines (SFMs) (Chen et al., 2017). Figure 2 shows experimental results on medical image sets.

As can be seen from Figure 2, SFMs are not superior to LUOPC since SFMs generally give sparse support vectors. It is found that LUOPC yields the best performance on these data sets since LUOPC explores the weight of each example in the set-valued objects. Among these methods, SFMs are sampling-based methods. SANP, RCRC and SMMs explore all possible representations of images. If the representations of images contain distorted features, these distorted features will affect the performance of classifiers. Our LUOPC makes use of the unbalanced optimal transport and reconstruction errors of data to achieve effective features. The experimental results indicate that it is reasonable to employ the optimal transport theory to classify set-valued data.

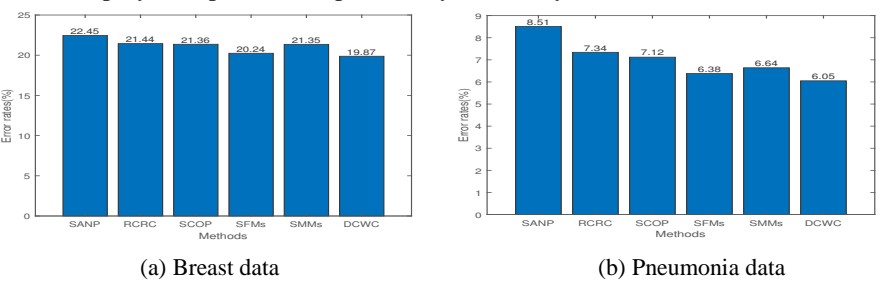

(a) Breast data  (b) Pneumonia data

Figure 2: Experimental results on medical image data sets.

## 5 CONCLUSIONS AND FURTHER WORK

We have introduced a feature learning framework relying on the local and unbalanced optimal transport and autoencoders. The use of the conditional probability in the proposed model facilitates to derive the optimization algorithm. The experimental results on real data sets demonstrate the feasibility of the proposed model. Our theoretical results depend on fully-connected neural networks, which may not be suitable for other types of neural networks. Since our proposed model contains autoencoders, how to select proper autoendcoders for data sets is worth exploring. In the future, we will focus on the problem of how to employ advanced autoencoders to improve our model to deal with complex data sets in the real world.

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

# A APPENDIX

## A.1 NOTATIONS

Table 4: Variables and notations used in our paper

| | |
|---|---|
| $\mathbb{X}$, $\mathbb{Y}$ | probability space with metric |
| $X$, $Y$ | random vectors |
| $\|a\|_r$ | $L_r$ norm of a vector |
| $\mu\,\nu$ | probability measure |
| $\rho(x, y)$ | the distance between $x \in R^m$ and $y \in R^m$ |
| $e_k$ | a $k$-dimensional vector whose elements are 1 |
| $\Pi()$ | joint measure |
| $p_{ij}$ | joint probability of $x_i$ and $y_j$ |
| $W^{\bar{r}}(\mu, \nu)$ | $\bar{r}-$order Wasserstein distance |
| $p_i,\ p_j$ | marginal probability |
| $q_i$ | prior marginal probability |
| $q_{i|j}$ | prior conditional probability |
| $p_{i|j}$ | conditional probability |
| $\{x_1, \cdots, x_n\}$ | sampled points from $X$ |
| $\delta_{x_i}$ | Dirac measure at $x_i$ |
| $f_\theta(x), g_\phi(y)$ | encoders |
| $\bar{f}_{\bar\theta}(z),\ \bar{g}_{\bar\phi}(z)$ | decoders |
| $y_{j|i}$ | k-neighbours of $x_i$ |
| $\sigma_{it}$ | the Rademacher random variable |
| $E_S$ | the expectation with respect to $S$ |
| $f_\theta(x_i)_t$ | the $t-th$ element of $f_\theta(x_i)$ |

## A.2 THE MODEL WITH THE F-DIVERGENCE

Let $f : R_+ \to R$ be a convex function with $f(1) = 0$. Let $p(x)$ and $q(x)$ be two probability density functions. The f-divergence between them is defined as: $D_f(p, q) = \int q(x) f(\frac{p(x)}{q(x)}) dx$. In terms of the f-divergence, our model can be written as

$$\min_{(\theta,\bar\theta,\phi,\bar\phi,p_{j|i},p_i)} \hat{L} := \sum\nolimits_{i=1}^n p_i \left( \sum\nolimits_{j=1}^k \rho(f_\theta(x_i), g_\phi(y_{j|i}))^{\bar{r}} p_{j|i} + \lambda_1 D_f(p_{\cdot|i}, q_{\cdot|i}) \right) + \lambda_2 D_f(p, q)$$

$$+ \lambda_3 \sum\nolimits_{i=1}^n \|x_i - \bar{f}_{\bar\theta} f_\theta(x_i)\|_2 p_i + \lambda_4 \sum\nolimits_{i,j=1}^{n,k} p_i p_{j|i} \|y_{j|i} - \bar{g}_{\bar\phi} g_\phi(y_{j|i})\|_2. \tag{13}$$

When $f(t) = t \ln(t)$, the f-divergence degenerates into the KL divergence. For the KL divergence, our model has good optimization properties, so we mainly discuss our model with the KL divergence.

### A.3 THE CONTINUOUS VERSION OF (4) IN OUR PAPER

Here, we give a continuous version of (4) which provides insights into understanding our discrete version. The following framework is used to extract effective features of data.

$$minL(\theta, \bar{\theta}, \phi, \bar{\phi}, p(y|x), p(x))) := \int \rho(f_\theta(x), g_\phi(y))^{\bar{r}} p(y|x)p(x)dxdy+$$

$$\lambda_1 \int p(x)\mathrm{KL}(p(y|x)||q(y|x))dx + \lambda_2\mathrm{KL}(p(x)||q(x))+ \tag{14}$$

$$\lambda_3 \int \|x - \bar{f}_{\bar{\theta}}f_\theta(x)\|_2 p(x)dx + \lambda_4 \int \|y - \bar{g}_{\bar{\phi}}g_\phi(y)\|_2 p(x)p(y|x)dxdy.$$

where $\mathrm{KL}(p(y|x)||q(y|x)) = \int p(y|x)\ln\frac{p(y|x)}{q(y|x)}dy$, $\mathrm{KL}(p(x)||q(x)) = \int p(x)\ln\frac{p(x)}{q(x)}dx$, $q(y|x)$ and $q(x)$ are prior (conditional) probabilities in the original space, and $\lambda_i(i = 1, \cdots, 4)$ are nonnegative hyperparameters. The first term in (14) denotes the transport cost in the embedding space. The second term is the Kullback-Leibler (KL) divergence to control conditional probabilities between $p(y|x)$ and $q(y|x)$. The third term is the KL divergence between $p(x)$ and $q(x)$. The last two terms involve the reconstruction errors of data $x$ and $y$. Trivial solutions of $\theta, \bar{\theta}, \phi, \bar{\phi}$ may be obtained if we do not employ the reconstruction error of data or the regularization terms for these parameters. From (14), we find that the loss function in the proposed model consists of the transport cost, reconstruction errors of data and additional regularization terms. The framework is generic since we do not give specific autoencoders and any transport cost can be used to replace $\rho()$. That is, we can employ some existing autoencoders to our framework.

### A.4 THE CLOSED-FORM SOLUTION TO THE PROBABILITY DISTRIBUTION

**Lemma 1**. *Let $\alpha \in R^n$ and $\beta \in R^n$ be two probability distributions. The following optimization problem has the closed-form solution.*

$$min_\alpha < \alpha, r > +\lambda KL(\alpha||\beta)$$

$$s.t. \sum_{i=1}^n \alpha_i = 1 \tag{15}$$

*where $\lambda$ is positive parameter and $r \in R^n$. The closed-form solution is denoted by $\alpha_i = \frac{\beta_i \exp\left(-\frac{r_i}{\lambda}\right)}{\sum_{i=1}^n \beta_i \exp\left(-\frac{r_i}{\lambda}\right)}$.*

The proof of Lemma 1 is given as follows. Note that the KL divergence is convex with respect to the first argument. In the specific condition(Melbourne, 2020), the KL divergence is strongly convex. Thus the optimization problem (15) is convex. Let us define the following Lagrange function by introducing Lagrange multiplier $v$.

$$L(\alpha, v) = <\alpha, r> +\lambda KL(\alpha||\beta) + v(\sum_{i=1}^n \alpha_i - 1) \tag{16}$$

By setting the derivative of $L(\alpha, v)$ with respect to $\alpha$ equal to zero, we have $r_i + \lambda(\log \alpha_i + 1 - \log \beta_i) + v = 0$. Thus we have $\alpha_i = \beta_i \exp\frac{-\lambda-v-r_i}{\lambda}$. Note that $\sum_{i=1}^n \alpha_i = 1$. This gives the solution to the optimization problem.

It is clear that using Lemma 1, we obtain the conditional probability $p_{j|i}$ and the marginal probability $p(i)$.

### A.5 THE PROOF OF THEOREM 1

**Lemma 2**. *For any $r \geq 1$ and two vectors $x$ and $y$ with proper dimensions, we have*

$$||x + y||_r \leq ||x||_r + ||y||_r. \tag{17}$$

**Lemma 3(Yin et al., 2022)**. *Let $x_1, \cdots, x_n$ be $n$ data points, and let $F$ be a class of vector-valued functions $f : X \longmapsto R^d$ and $h_i: R^d \longmapsto R$ be functions with the Lipschitz constant $\ell$. Then we*

*have*

$$E \sup_{f \in F} \sum_{i=1}^{n} \sigma_i h_i(f(x_i)) \leq \sqrt{2} \ell E \sup_{f \in F} \sum_{i=1}^{n} \sum_{j=1}^{d} \sigma_{ij}(f_i)_j, \tag{18}$$

*where $\sigma_{ij}$ is an independent doubly indexed Rademacher sequence and $(f_i)_j$ is the $j$-th component of $f(x_i)$.*

**Lemma 4** (Kuritsyn, 1986). *Let $a$ be a vector containing $m$ elements. The following Khintchine inequality holds*

$$A_r (\sum_{i=1}^{m} a_i^2)^{\frac{1}{2}} \leq (E|\sum_{i=1}^{m} \sigma_i a_i|^r)^{\frac{1}{r}} \leq B_r (\sum_{i=1}^{m} a_i^2)^{\frac{1}{2}}, \tag{19}$$

*where $A_r$ and $B_r$ are constants depending on $r$. When $r = 1$, we have $B_r = 1$.*

To give the generalization bound of $\hat{L}_S(\theta, \bar{\theta})$, we rewrite $\hat{L}_S(\theta, \bar{\theta})$ by removing $p_{j|i}$. Thus, $\hat{L}_S(\theta, \bar{\theta})$ can be formulated as

$$\hat{L}_S(\theta, \bar{\theta}) = -\frac{\lambda_1}{n} \sum_{i=1}^{n} \ln \sum_{j=1}^{k} q_{j|i} \exp(-\rho(f_\theta(x_i) - f_\theta(y_{j|i}))^{\bar{r}}/\lambda_1) + \sum_{i=1}^{n} \frac{\lambda_3}{n} \|x_i - \bar{f}_{\bar{\theta}} f_\theta(x_i)\|_2. \tag{20}$$

Let $S'$ be the data set where only a data point is different from the data set $S$, e.g., $\bar{x}_s$. Let $\hat{L}_S(\theta, \bar{\theta})$ denote the empirical loss from $S'$. Let us define the following functions:

$$\psi_S = \sup_{\theta, \bar{\theta}} (L(\theta, \bar{\theta}) - \hat{L}_S(\theta, \bar{\theta})), \tag{21}$$

$$\psi'_S = \sup_{\theta, \bar{\theta}} (L(\theta, \bar{\theta}) - \hat{L}_{S'}(\theta, \bar{\theta})). \tag{22}$$

From (21) and (22), we have

$$|\psi_S - \psi'_S| \overset{(1)}{\leq} |\hat{L}_S(\theta, \bar{\theta}) - \hat{L}_{S'}(\theta, \bar{\theta})| = \frac{1}{n} \sup_{\theta, \bar{\theta}} | - \lambda_1 \ln \sum_{j=1}^{k} q_{j|i} \exp(-\rho(f_\theta(x_s) - f_\theta(y_{j|s}))^{\bar{r}}/\lambda_1)$$

$$+ \lambda_3 \ln \sum_{j=1}^{k} q_{j|i} \exp(-\rho(f_\theta(\bar{x}_s) - f_\theta(y_{j|s}))^{\bar{r}}/\lambda_1) + \lambda_3 \|x_s - \bar{f}_{\bar{\theta}} f_\theta(x_s)\|_2 - \lambda_3 \|\bar{x}_s - \bar{f}_{\bar{\theta}} f_\theta(\bar{x}_s)\|_2 |$$

$$\leq \frac{1}{n} \{ \sup_{\theta} \lambda_1 | \ln \sum_{j=1}^{k} q_{j|i} \exp(-\rho(f_\theta(x_s) - f_\theta(y_{j|s}))^{\bar{r}}| + \sup_{\theta} |\lambda_1 \ln \sum_{j=1}^{k} q_{j|i} \exp(-\rho(f_\theta(\bar{x}_s) - f_\theta(y_{j|s}))^{\bar{r}}|$$

$$+ \sup_{\theta, \bar{\theta}} \lambda_3 \|x_s - \bar{f}_{\bar{\theta}} f_\theta(x_s)\|_2 + \sup_{\theta, \bar{\theta}} \lambda_3 \|\bar{x}_s - \bar{f}_{\bar{\theta}} f_\theta(\bar{x}_s)\|_2 \}. \tag{23}$$

From assumptions of A0 in our paper, we have

$$\rho(f_\theta(x_s) - f_\theta(y_{j|s}))^{\bar{r}} = \varphi(f_\theta(x_s) - f_\theta(y_{j|s}))^{\bar{r}}. \tag{24}$$

Note that the function $\varphi()$ is Lipschitz continuous and its Lipschitz constant is $\ell$. Hence, we have

$$\varphi(f_\theta(x_s) - f_\theta(y_{j|s})) \leq \ell \|f_\theta(x_s) - f_\theta(y_{j|s})\|_2. \tag{25}$$

From $\|f_\theta(x_t)\|_2 \leq M$ and $\|f_\theta(y_{j|t})\|_2 \leq M$, we have

$$\|f_\theta(x_s) - f_\theta(y_{j|s})\|_2 \leq 2M. \tag{26}$$

Similarly, we have

$$\|x_s - \bar{f}_{\bar{\theta}} f_\theta(x_s)\|_2 \leq 2M. \tag{27}$$

From (23), (25), and (27), we have

$$
|\psi_S - \psi'_S| \leq \frac{1}{n}\{|\lambda_1 \ln \sum_{j=1}^{k} q_{j|i} \exp(-(2M\ell)^{\bar{r}}/\lambda_1)|
$$
$$
+ |\lambda_1 \ln \sum_{j=1}^{k} q_{j|i} \exp(-(2M\ell)^{\bar{r}}/\lambda_1)| + 2\lambda_3 2M\} \leq \frac{2(2ML)^{\bar{r}} + 4\lambda_3 M}{n}. \tag{28}
$$

In the following, we consider the expectation of $\psi_S$ with respect to the data set $S$, denoted by $E_S(\psi_S)$:

$$
E_S((\psi_S)) = E_S(\sup_{\theta,\bar{\theta}}(L(\theta,\bar{\theta}) - \hat{L}_S(\theta,\bar{\theta})))
$$

$$
\overset{(1)}{\leq} 2E_{S,\sigma}\frac{1}{n}\sup_{\theta,\bar{\theta}}\{\sum_{i=1}^{n} -\lambda_1\sigma_i \ln \sum_{j=1}^{k} q_{j|i}\exp(-\rho(f_\theta(x_i) - f_\theta(y_{j|i}))^{\bar{r}}/\lambda_1) + \lambda_3\sum_{i=1}^{n}\sigma_i\|x_i - \bar{f}_{\bar{\theta}}f_\theta(x_i)\|_2\}
$$

$$
\overset{(2)}{\leq} 2E_{S,\sigma}\frac{1}{n}\sup_{\theta,\bar{\theta}}\{\sum_{i=1}^{n}\sigma_i\sum_{j=1}^{k} q_{j|i}\rho(f_\theta(x_i) - f_\theta(y_{j|i}))^{\bar{r}} + \lambda_3\sum_{i=1}^{n}\sigma_i\|x_i - \bar{f}_{\bar{\theta}}f_\theta(x_i)\|_2\}
$$

$$
\overset{(3)}{\leq} 2E_{S,\sigma}\frac{1}{n}\sup_{\theta}\sum_{i=1}^{n}\sigma_i\sum_{j=1}^{k} q_{j|i}\rho(f_\theta(x_i) - f_\theta(y_{j|i}))^{\bar{r}} + 2E_{S,\sigma}\frac{1}{n}\sup_{\theta,\bar{\theta}}\lambda_3\sum_{i=1}^{n}\sigma_i\|x_i - \bar{f}_{\bar{\theta}}f_\theta(x_i)\|_2. \tag{29}
$$

In (29), the first inequality comes from the symmetrization of random variables, and the second inequality uses Jensen inequality from the fact that $-lnx$ is a convex function. Note that the Lipschitz constant of the norm $\|\cdot\|$ is 1. The function $\|x\|_r^r$ is not Lipschitz continuous if the variable $x$ takes the infinite values. However, with the assumptions we provide, there exists the constant $M_1 = \bar{r}(2M\ell)^{\bar{r}-1}l$ such that $\rho(f_\theta(x_i) - f_\theta(y_{j|i}))^{\bar{r}}$ is Lipschitz continuous. Using Lemma 3, we have

$$
\begin{aligned}
E_S(\psi_S) &\overset{(1)}{\leq} 2E_{S,\sigma}\frac{1}{n}\sup_{\theta,\bar{\theta}}\sum_{i=1}^{n}\sum_{t=1}^{d}\sigma_{it}\sum_{j=1}^{k}q_{j|i}\sqrt{2}M_1(f_\theta(x_i)-f_\theta(y_{j|i})_t) \\
&+ 2E_{S,\sigma}\frac{1}{n}\sup_{\theta,\bar{\theta}}\lambda_3\sum_{i=1}^{n}\sum_{t=1}^{m}\sqrt{2}\sigma_{it}(\bar{f}_{\bar{\theta}}f_\theta(x_i))_t \\
&\overset{(2)}{\leq} 2E_{S,\sigma}\frac{1}{n}\sup_{\theta}\sum_{i=1}^{n}\sum_{t=1}^{d}\sigma_{it}\sum_{j=1}^{k}q_{j|i}\sqrt{2}M_1(f_\theta(x_i))_t \\
&+ 2E_{S,\sigma}\frac{1}{n}\sup_{\theta}\sum_{i=1}^{n}\sum_{t=1}^{d}\sigma_{it}\sum_{j=1}^{k}q_{j|i}\sqrt{2}M_1(-f_\theta(y_{j|i})_t) \\
&+ 2E_{S,\sigma}\frac{1}{n}\sup_{\theta,\bar{\theta}}\lambda_3\sum_{i=1}^{n}\sum_{t=1}^{m}\sqrt{2}\sigma_{it}(\bar{f}_{\bar{\theta}}f_\theta(x_i))_t \\
&\overset{(3)}{\leq} 2E_{S,\sigma}\frac{1}{n}\sup_{\theta}\sum_{i=1}^{n}\sum_{t=1}^{d}\sigma_{it}\sqrt{2}M_1(f_\theta(x_i))_t + 2E_{S,\sigma}\frac{1}{n}\sup_{\theta}\sum_{i=1}^{n}\sum_{t=1}^{d}\sigma_{it}\sum_{j=1}^{k}q_{j|i}\sqrt{2}M_1(-f_\theta(y_{j|i})_t) \\
&+ 2E_{S,\sigma}\frac{1}{n}\sup_{\theta,\bar{\theta}}\lambda_3\sum_{i=1}^{n}\sum_{t=1}^{m}\sqrt{2}\sigma_{it}(\bar{f}_{\bar{\theta}}f_\theta(x_i))_t \\
&\overset{(4)}{\leq} 2E_{S,\sigma}\frac{1}{n}\sup_{\theta}\sum_{i=1}^{n}\sum_{t=1}^{d}\sigma_{it}\sqrt{2}M_1(f_\theta(x_i))_t + 2E_{S,\sigma}\frac{1}{n}\sup_{\theta}\sum_{i=1}^{n}\sum_{t=1}^{d}\sigma_{it}\sum_{j=1}^{k}q_{j|i}\sqrt{2}M_1(-f_\theta(x_i)_t) \\
&+ 2E_{S,\sigma}\frac{1}{n}\sup_{\theta,\bar{\theta}}\lambda_3\sum_{i=1}^{n}\sum_{t=1}^{m}\sqrt{2}\sigma_{it}(\bar{f}_{\bar{\theta}}f_\theta(x_i))_t.
\end{aligned}
\tag{30}
$$

In (30), the first inequality uses Lemma 4, and the second inequality uses the property of $sup$. The third inequality uses the fact that $\sum_{j=1}^{k}q_{j|i}=1$. Since $y_{j|i}$ depends on $x_i$, we assume that $y_{j|i}$ is an independent copy of $x_i$. Hence, the fourth inequality is to replace $y_{j|i}$ with $x_i$. From (30), we have

$$
\begin{aligned}
E_S(\psi_S) &\leq 2E_{S,\sigma}\frac{1}{n}\sup_{\theta}\sum_{t=1}^{d}|\sum_{i=1}^{n}\sigma_{it}\sqrt{2}M_1(f_\theta(x_i))_t| + 2E_{S,\sigma}\frac{1}{n}\sup_{\theta}\sum_{t=1}^{d}|\sum_{i=1}^{n}\sigma_{it}\sqrt{2}M_1(-f_\theta(x_i)_t)| \\
&+ 2E_{S,\sigma}\frac{1}{n}\sup_{\theta,\bar{\theta}}\lambda_3\sum_{t=1}^{m}|\sum_{i=1}^{n}\sqrt{2}\sigma_{it}(\bar{f}_{\bar{\theta}}f_\theta(x_i))_t| \leq 4\sqrt{2}M_1 E_{S,\sigma}\frac{1}{n}\sup_{\theta}\sum_{t=1}^{d}|\sum_{i=1}^{n}\sigma_i(f_\theta(x_i))_t| \\
&+ 2\sqrt{2}E_{S,\sigma}\frac{1}{n}\sup_{\theta,\bar{\theta}}\lambda_3\sum_{t=1}^{m}|\sum_{i=1}^{n}\sigma_i(\bar{f}_{\bar{\theta}}f_\theta(x_i))_t|.
\end{aligned}
\tag{31}
$$

From (31), we find that the upper bound of $E_S(\psi_S)$ depends on the Rademacher complexity of the encoders and decoders. From (28) and (31), we obtain that with probability at least $1-\tau$, the following inequality holds for $\theta$ and $\bar{\theta}$ in proper parameter spaces by using McDiarmid inequality:

$$
\hat{L}_S(\theta,\bar{\theta}) \leq L(\theta,\bar{\theta}) + 4\sqrt{2}M_1 R_1 + 2\sqrt{2}R_2 + \chi_1\sqrt{\frac{-log\tau}{2n}}
\tag{32}
$$

where $\chi_1 = \frac{2(2M)^{\bar{r}}+4\lambda_3 M}{n}$, $R_1 = E_{S,\sigma}\frac{1}{n}\sup_{\theta}\sum_{t=1}^{d}|\sum_{i=1}^{n}\sigma_{it}(f_\theta(x_i))_t|$, and $R_2 = E_{S,\sigma}\frac{1}{n}\sup_{\theta,\bar{\theta}}\lambda_3\sum_{t=1}^{m}|\sum_{i=1}^{n}\sigma_{it}(\bar{f}_{\bar{\theta}}f_\theta(x_i))_t|$.

### A.6 OPTIMIZATION OF (10) IN OUR PAPER

In the following, we use the alternative optimization method to solve the optimization model:

$$
min_{(\theta,\bar{\theta},p_{j|i},p_i,z_j)}\hat{L} := \sum_{i,j=1}^{n,k} p_i\rho(f_\theta(x_i),z_j)^{\bar{r}}p_{j|i}+
$$
$$
\lambda_1\sum_{i=1}^n p_i\mathrm{KL}(p_{\cdot|i}||q_{\cdot|i}) + \sum_{i=1}^n \lambda_3 p_i\|x_i - \bar{f}_{\bar{\theta}}f_\theta(x_i)\|_2 + \lambda_2\mathrm{KL}(p||q). \tag{33}
$$

(a): Update $p_{j|i}$ by fixing other variables. When we fix $\theta,\bar{\theta}, z_j$ and $p_i$, we solve the following model:

$$
min_{p_{j|i}} \sum_{i,j=1}^{n,k} \rho(f_\theta(x_i),z_j)^{\bar{r}}p_i p_{j|i} + \lambda_1\sum_{i=1}^n p_i\mathrm{KL}(p_{\cdot|i}||q_{\cdot|i}). \tag{34}
$$

It is noted that (34) is a strongly convex optimization problem. It is of interest to note that it has a closed-form solution, denoted by

$$
p_{j|i} = \frac{q_{j|i}exp(-L_{j|i}^{op}/\lambda_1)}{\sum_{j=1}^k q_{j|i}exp(-L_{j|i}^{op}/\lambda_1)}, \tag{35}
$$

where $L_{j|i}^{op} = \rho(f_\theta(x_i),z_j)^{\bar{r}}$.

(b): Update $p_i$ by fixing other variables. Given $\theta,\bar{\theta},z_j$, and $p_{j|i}$, we achieve $p_i$ by solving the following problem:

$$
min_{p_i} \sum_{i,j=1}^{n,k} \rho(f_\theta(x_i),z_j)^{\bar{r}}p_i p_{j|i} + \lambda_1\sum_{i=1}^n p_i\mathrm{KL}(p_{\cdot|i}||q_{\cdot|i}) + \lambda_2\mathrm{KL}(p||q)+
$$
$$
\lambda_3\sum_{i=1}^n \|x_i - \bar{f}_{\bar{\theta}}f_\theta(x_i)\|_2 p_i. \tag{36}
$$

It is observed that the objective function in (36) is strongly convex. Thus, there exists a unique solution to $p_i$. The closed-form solution is denoted by

$$
p_i = \frac{q_i exp(-(L_i^{op} + L_i^{enre})/\lambda_2)}{\sum_{i=1}^n q_i exp(-(L_i^{op} + L_i^{enre})/\lambda_2)}, \tag{37}
$$

where $L_i^{op} = \sum_{j=1}^k \rho(f_\theta(x_i),z_j)^{\bar{r}}p_{j|i}$ and $L_i^{enre} = \lambda_3\|x_i - \bar{f}_{\bar{\theta}}f_\theta(x_i)\|_2 + \mathrm{KL}(p_{\cdot|i}||q_{\cdot|i})$.

(c): Update $z_j$ by fixing other variables. If $\rho$ takes the Euclidean distance and $\bar{r} = 2$, $z_j$ has the following solution:

$$
z_j = \frac{\sum_{i=1}^n q_i q_{j|i}f_\theta(x_i)}{\sum_{i=1}^n q_i q_{j|i}}. \tag{38}
$$

(d): Update $\theta$ and $\bar{\theta}$ by fixing other variables. In this step, we try to learn the parameters of autoencoders. Specifically, we solve the following optimization problem:

$$
min_{(\theta,\bar{\theta})} \sum_{i,j=1}^{n,k} \rho(f_\theta(x_i),z_j)^{\bar{r}}p_i p_{j|i} + \lambda_3\sum_{i=1}^n p_i\|x_i - \bar{f}_{\bar{\theta}}f_\theta(x_i)\|_2. \tag{39}
$$

Note that the objective function in (39) is nonconvex. We cannot obtain the global optimal solution. We generally update these parameters of models through the chain rule in the framework of neural networks. In this work, we resort to automatic differentiation to learn these parameters. For the sake of completeness, we summarize the main steps of solving (33) in Algorithm 2.

### A.7 THE PROOF OF THEOREM 2

We obtain $\hat{L}_S^c(\theta,\bar{\theta},z_j)$ by removing $P_{j|i}$. Thus, $\hat{L}_S^c(\theta,\bar{\theta},z_j)$ can be formulated as

$$
\hat{L}_S^c(\theta,\bar{\theta},z_j) = \frac{-1}{n}\sum_{i=1}^n \lambda_1\ln\sum_{j=1}^k q_{j|i}\exp(-\rho(f_\theta(x_i),z_j)^{\bar{r}}/\lambda_1) + \sum_{i=1}^n \frac{\lambda_3}{n}\|x_i - \bar{f}_{\bar{\theta}}f_\theta(x_i)\|_2. \tag{40}
$$

| Algorithm 2: Optimization algorithm to (33) |
| --- |
| 1: Given $\lambda_i, q_{j\|i}, q_i$, and initialize $p_i = q_i, p_{j\|i} = q_{j\|i}$ |
| 2: **For** t=1 to T **do** |
|    2.1: solve (39) to achieve the parameters $(\theta, \bar{\theta})$; |
|    2.2: solve (34) to achieve $p_{j\|i}$; |
|    2.3: solve (36) to achieve $p_i$; |
|    2.4: use (38) to achieve $z_j$; |
| 3: Output: the encoders and decoders. |

Let $S'$ be the data set where only a data point is different from the data set $S$, e.g., $\bar{x}_s$. Let $\hat{L}_S^c(\theta, \bar{\theta})$ denote the empirical loss from $S'$. Let us define the following functions:

$$\psi_S = \sup_{\theta, \bar{\theta}, z_j} (L(\theta, \bar{\theta}) - \hat{L}_S^c(\theta, \bar{\theta})), \tag{41}$$

$$\psi_S' = \sup_{\theta, \bar{\theta}, z_j} (L(\theta, \bar{\theta}) - \hat{L}_{S'}^c(\theta, \bar{\theta})). \tag{42}$$

From (41) and (42), we have

$$|\psi_S - \psi_S'| \overset{(1)}{\leq} |\hat{L}_S^c(\theta, \bar{\theta}) - \hat{L}_{S'}^c(\theta, \bar{\theta})|$$

$$= \frac{1}{n} \sup_{\theta, \bar{\theta}, z_j} |-\lambda_1 \ln \sum_{j=1}^k q_{j|i} \exp(-\rho(f_\theta(x_s), z_j)^{\bar{r}}/\lambda_1) + \lambda_1 \ln \sum_{j=1}^k q_{j|i} \exp(-\rho(f_\theta(\bar{x}_s), z_j))^{\bar{r}}/\lambda_1)$$

$$+ \lambda_3 \|x_s - \bar{f}_{\bar{\theta}} f_\theta(x_s)\|_2 - \lambda_3 \|\bar{x}_s - \bar{f}_{\bar{\theta}} f_\theta(\bar{x}_s)\|_2|$$

$$\leq \frac{1}{n} \{\sup_{\theta, y_j} |\lambda_1 \ln \sum_{j=1}^k q_{j|i} \exp(-\rho(f_\theta(x_s), z_j)^{\bar{r}}/\lambda_1)| + \sup_{\theta, z_j} |\lambda_1 \ln \sum_{j=1}^k q_{j|i} \exp(-\rho(f_\theta(x_s), z_j)^{\bar{r}}/\lambda_1)|$$

$$+ \sup_{\theta, \bar{\theta}} \lambda_3 \|x_s - \bar{f}_{\bar{\theta}} f_\theta(x_s)\|_2 + \sup_{\theta, \bar{\theta}} \lambda_3 \|\bar{x}_s - \bar{f}_{\bar{\theta}} f_\theta(\bar{x}_s)\|_2\}. \tag{43}$$

Using the assumption of A0 in our paper, we have

$$\rho(f_\theta(x_s), z_j)^{\bar{r}} = \varphi(f_\theta(x_s) - z_j)^{\bar{r}}. \tag{44}$$

From $\|f_\theta(x_t)\| \leq M$ and $\|z_j\| \leq M$, we have

$$\rho(f_\theta(x_s), z_j)^{\bar{r}} \leq (2M\ell)^{\bar{r}}. \tag{45}$$

Similarly, we have

$$\|x_s - \bar{f}_{\bar{\theta}} f_\theta(x_s)\|_2 \leq 2M. \tag{46}$$

Thus, (43) leads to

$$|\psi_S - \psi_S'| \leq \frac{1}{n} \{|\lambda_1 \ln \sum_{j=1}^k q_{j|i} \exp(-(2M\ell)^{\bar{r}}/\lambda_1)|$$

$$+ |\lambda_1 \ln \sum_{j=1}^k q_{j|i} \exp(-(2M\ell)^{\bar{r}}/\lambda_1)| + 2\lambda_3 2M\}$$

$$\leq \frac{2(2M\ell)^{\bar{r}} + 4\lambda_3 M}{n}. \tag{47}$$

In the following, we consider the expectation of $\psi_S$ with respect to the data set $S$, denoted by $E_S(\psi_S)$

$$E_S(\psi_S) = E_S\big(\sup_{\theta,\bar{\theta},z_j} (L(\theta,\bar{\theta},z_j) - \hat{L}_S^c(\theta,\bar{\theta},z_j))\big)$$

$$\overset{(1)}{\leq} 2E_{S,\sigma}\frac{1}{n}\sup_{\theta,\bar{\theta},z_j} \{\sum_{i=1}^n -\sigma_i\lambda_1 \ln\sum_{j=1}^k q_{j|i}\exp(-\rho(f_\theta(x_i),z_j)^{\bar{r}}/\lambda_1) + \lambda_3\sum_{i=1}^n \sigma_i\|x_i - \bar{f}_{\bar{\theta}}f_\theta(x_i)\|_2\}$$

$$\overset{(2)}{\leq} 2E_{S,\sigma}\frac{1}{n}\sup_{\theta,\bar{\theta},z_j} \{\sum_{i=1}^n \sigma_i\sum_{j=1}^k q_{j|i}\rho(f_\theta(x_i),z_j)^{\bar{r}} + \lambda_3\sum_{i=1}^n \sigma_i\|x_i - \bar{f}_{\bar{\theta}}f_\theta(x_i)\|_2\}$$

$$\overset{(3)}{\leq} 2E_{S,\sigma}\frac{1}{n}\sup_{\theta,\bar{\theta},z_j} \sum_{i=1}^n \sigma_i\sum_{j=1}^k q_{j|i}\rho(f_\theta(x_i),z_j)^{\bar{r}} + 2E_{S,\sigma}\frac{1}{n}\sup_{\theta,\bar{\theta}}\lambda_3\sum_{i=1}^n \sigma_i\|x_i - \bar{f}_{\bar{\theta}}f_\theta(x_i)\|_2.$$

$$(48)$$

In (48), the first inequality comes from the symmetrization of random variables, and the second inequality uses Jensen inequality from the fact that $-lnx$ is a convex function. Note that the Lipschitz constant of the norm $\|\cdot\|$ is 1, but the function $\|x\|_r^r$ is not Lipschitz if the variable $x$ takes the infinite values. However, with the assumptions we provide, there exists the constant $M_1$ such that $\rho(f_\theta(x_i),y_j)^{\bar{r}}$ is Lipschitz continuous. Using Lemma 3, we have

$$E_S(\psi_S) \overset{(1)}{\leq} 2E_{S,\sigma}\frac{1}{n}\sup_{\theta,z_j}\sum_{i=1}^n\sum_{t=1}^d \sigma_{it}\sum_{j=1}^k q_{j|i}\sqrt{2}M_1(f_\theta(x_i) - z_j)_t$$

$$+ 2E_{S,\sigma}\frac{1}{n}\sup_{\theta,\bar{\theta}}\lambda_3\sum_{i=1}^n\sum_{t=1}^m \sqrt{2}\sigma_{it}(\bar{f}_{\bar{\theta}}f_\theta(x_i))_t$$

$$\overset{(2)}{\leq} 2E_{S,\sigma}\frac{1}{n}\sup_{\theta}\sum_{i=1}^n\sum_{t=1}^d \sigma_{it}\sum_{j=1}^k q_{j|i}\sqrt{2}M_1(f_\theta(x_i))_t + 2E_{S,\sigma}\frac{1}{n}\sup_{z_j}\sum_{i=1}^n\sum_{t=1}^d \sigma_{it}\sum_{j=1}^k q_{j|i}\sqrt{2}M_1(-z_j)_t$$

$$+ 2E_{S,\sigma}\frac{1}{n}\sup_{\theta,\bar{\theta}}\lambda_3\sum_{i=1}^n\sum_{t=1}^m \sqrt{2}\sigma_{it}(\bar{f}_{\bar{\theta}}f_\theta(x_i))_t$$

$$\overset{(3)}{\leq} 2E_{S,\sigma}\frac{1}{n}\sup_{\theta}\sum_{i=1}^n\sum_{t=1}^d \sigma_{it}\sqrt{2}M_1(f_\theta(x_i))_t + 2E_{S,\sigma}\frac{1}{n}\sup_{z_j}\sum_{i=1}^n\sum_{t=1}^d \sigma_{it}\sum_{j=1}^k q_{j|i}\sqrt{2}M_1(-z_j)_t$$

$$+ 2E_{S,\sigma}\frac{1}{n}\sup_{\theta,\bar{\theta}}\lambda_3\sum_{i=1}^n\sum_{t=1}^m \sqrt{2}\sigma_{it}(\bar{f}_{\bar{\theta}}f_\theta(x_i))_t$$

$$(49)$$

In (49), the first inequality comes from Lemma 2, and the second inequality uses the property of $sup$. The third inequality uses the fact that $\sum_{j=1}^k q_{j|i} = 1$. From (49), we have

$$E_S(\psi_S) \leq 2E_{S,\sigma}\frac{1}{n}\sup_{\theta}\sum_{t=1}^d |\sum_{i=1}^n \sigma_{it}\sqrt{2}M_1(f_\theta(x_i))_t|$$

$$+ 2E_{S,\sigma}\frac{1}{n}\sup_{z_j}\sum_{t=1}^d\sum_{j=1}^k |(z_j)_t||\sum_{i=1}^n \sigma_{it}\sqrt{2}M_1 q_{j|i}| + 2E_{S,\sigma}\frac{1}{n}\sup_{\theta,\bar{\theta}}\lambda_3\sum_{t=1}^m |\sum_{i=1}^n \sqrt{2}\sigma_{it}(\bar{f}_{\bar{\theta}}f_\theta(x_i))_t|$$

$$\leq 2\sqrt{2}M_1 E_{S,\sigma}\frac{1}{n}\sup_{\theta}\sum_{t=1}^d |\sum_{i=1}^n \sigma_{it}(f_\theta(x_i))_t| + 2\sqrt{2}M_1 M E_\sigma\frac{1}{n}\sum_{t=1}^d\sum_{j=1}^k |\sum_{i=1}^n \sigma_{it}q_{j|i}|$$

$$+ 2E_{S,\sigma}\frac{1}{n}\sup_{\theta,\bar{\theta}}\lambda_3\sum_{t=1}^m |\sum_{i=1}^n \sqrt{2}\sigma_{it}(\bar{f}_{\bar{\theta}}f_\theta(x_i))_t|.$$

$$(50)$$

Further we have

$$E_S(\psi_S) \leq 2\sqrt{2}M_1 E_{S,\sigma} \frac{1}{n} \sup_\theta \sum_{t=1}^d |\sum_{i=1}^n \sigma_{it}(f_\theta(x_i))_t| + \frac{2\sqrt{2}M_1 M dk}{\sqrt{n}}$$
$$+ 2E_{S,\sigma} \frac{1}{n} \sup_{\theta,\bar\theta} \lambda_3 \sum_{t=1}^m |\sum_{i=1}^n \sqrt{2}\sigma_{it}(\bar{f}_{\bar\theta} f_\theta(x_i))_t|. \tag{51}$$

In (51), we use Lemma 4 to obtain the last inequality. From (51), we find that the upper bound of $E_S(\psi_S)$ depends on the number of anchors, and Rademacher complexities of the encoders and decoders. From (47) and (51), we obtain that with probability at least $1-\tau$, the following inequality holds for $\theta$, $\bar\theta$, and $z_j$ in proper parameter spaces by using McDiarmid inequality:

$$\hat{L}_S^c(\theta, \bar\theta, z_j) \leq L(\theta, \bar\theta, z_j) + 2\sqrt{2}M_1 R_1 + 2\sqrt{2}R_2 + \frac{\chi_1 + \chi_2}{\sqrt{n}} \tag{52}$$

where $\chi_1 = \frac{2(2M)^r + 4\lambda_3 M}{n}\sqrt{\frac{-ln\tau}{2}}$, $\chi_2 = 2\sqrt{2}M_1 M dk$, $R_1 = E_{S,\sigma}\frac{1}{n}\sup_\theta \sum_{t=1}^d |\sum_{i=1}^n \sigma_{it}(f_\theta(x_i))_t|$, and $R_2 = E_{S,\sigma}\frac{1}{n}\sup_{\theta,\bar\theta} \lambda_3 \sum_{t=1}^m |\sum_{i=1}^n \sigma_{it}(\bar{f}_{\bar\theta} f_\theta(x_i))_t|$.

## A.8 EXPERIMENTAL SETTING AND ADDITIONAL EXPERIMENTS FOR OUR MODEL

All experiments are conducted on a PC with an Intel Core i7- CPU and a RTX 3080 GPU. The structure of encoders we use in this paper is a fully-connected network with the form of $[m, 500, 500, 2000, d]$ and the decoder is a mirror of the encoder, where $m$ is the dimension of input data and $d$ is the dimension of the latent space. The popular ReLU functions are employed in each layer. We employ Adam (Kingma & Ba, 2015) as the backpropagation optimizer. In the experiments, $\rho$ takes the L2 norm, $\bar{r} = 2$, and $\lambda_2 = 1000$. We let $\lambda_4 = 0$ due to the use of the same encoders. The parameters $\lambda_1$ and $\lambda_3$ are selected from the set $\{10^i, i = -3, -2, \cdots, 2, 3\}$. In the classification experiments, we need to determine $k$ in (4) in our paper. Note that $x_1, \cdots, x_n$ consist of the training set and $y_{j|i}(j = 1, \cdots, k)$ are taken from the samples that have the same label as $x_i$. We think that the samples in the same class are neighbors. Thus, $k$ will be determined by the number of samples in each class. As a result, $k$ will vary since the number of samples in each class is different in the training set. In the clustering experiments, $k$ in the model of (10) is set to be the number of clusters. We find that good performance can be obtained by this setting since the encoder has strong representations of features.

The outer loop is 10 iterations and the inner loop for autoencoders is run with an Adam optimizer for 100 epochs, with an initial learning rate of 0.001. In the data sets except for two large-scale data sets , all data sets are handled with a full-batch mode. For the large-scale data sets, the batch size is 2000.

The data sets from the UCI repository are Dna (180 attributes /3 classes /2000 samples), Pendigits (16/10/7494), Satimage (36/6/4435), Iris(4/3/150), and WaveForm (21/3/5000). In addition, we also explore four face image data sets and an object data set. The ORL face database contains 40 distinct persons and each person has taken 10 different images. The UMIST face database contains 564 face images of 20 distinct subjects. The Yale face database contains 165 images of 15 individuals. The COIL database contains 1440 images with black background of 20 objects. The MSRA face data set consists of 1799 images of 12 subjects. All the images are normalized to a resolution of $32 \times 32$ pixels for computational efficiency. For each data set from the UCI repository, we randomly choose fifty percent samples to form the training set and the rest is used as the test set. The performance of each model is evaluated over twenty random splits of each data set. The additional five runs are employed to select the parameters of each model.

The MNIST data set contains 60,000 training samples and 10,000 test samples, and the FashionM-NIST data set has 60,000 training samples and 10,000 test samples. The dimension of samples in these two data sets is 784.

For set-valued objects, we employ the features extracted from a pre-trained convolutional neural network (CNN), i.e., ReNet18, and the features are taken from the layer of res5b-relu. The extracted

features of each image have a tensor representation of $7 \times 7 \times 512$ dimensions. To reduce the computational cost, we pre-process the features of each image. That is, we perform the mean operation along the first axis and downsample the features with a factor of 4 along the third axis. Thus, we obtain the features whose dimensions are $7 \times 128$. Namely, each image can be regarded as a set-valued object containing seven examples with 128 dimensions. We randomly choose $50\%$ of the samples as the training set, $30\%$ of the samples as the validation set, and the other images as the testing set. Experimental results are averaged over 10 runs. Table 5 shows the clustering accuracy of various methods on small data sets, and Table 6 shows the adjusted rand index(ARI) of various methods on small data sets. Table 7 shows clustering accuracy and the adjusted rand index(ARI) of various methods on two large-scale data sets. Table 8 lists the running time of various methods using RTX 3080. When the deep neural networks are used, we list the running time of algorithms in terms of each epoch.

Table 5: Clustering Accuracy $(\%)$ of various methods and their standard deviations on data sets

| data sets | KKM | KFKM | KPKM | DEC | IDEC | DFKM | AE | AEL | LUOPSC |
|---|---|---|---|---|---|---|---|---|---|
| Dna | 60.71±1.34 | 62.59±2.25 | 63.42±2.62 | 79.13±2.54 | 79.95±2.62 | 77.78±2.36 | 76.36±2.05 | 79.25±1.32 | **80.52±1.45** |
| Pendigits | 67.15±2.38 | 67.87±1.65 | 68.33±2.67 | 69.72±1.98 | 70.39±2.39 | 71.26±1.76 | 68.46±1.92 | 71.68±2.005 | **72.34±1.62** |
| Iris | 85.32±2.13 | 86.22±2.16 | 87.34±2.19 | 88.52±2.46 | 89.34±2.63 | 80.50±2.63 | 88.31±2.06 | 89.96±1.90 | **90.56±1.88** |
| Satimage | 73.55±2.71 | 74.62±3.06 | 75.46±3.03 | 77.32±3.41 | 79.41±3.08 | 78.26±3.25 | 75.93±3.46 | **79.92±3.12** | 78.34±2.76 |
| Waveform | 50.96±1.89 | 51.39±1.72 | 54.26±2.37 | 56.38±1.70 | 57.30±2.42 | 57.52±2.26 | 54.82±2.15 | 57.4±2.32 | **56.93±2.08** |
| ORL | 53.46±1.26 | 52.57±3.11 | 54.75±3.06 | 56.88±2.42 | 57.34±2.26 | 55.49±2.13 | 55.42±2.31 | 57.30±2.14 | **58.19±2.06** |
| Yale | 66.35±3.61 | 66.27±3.72 | 64.32±3.84 | 65.63±4.06 | 66.70±3.42 | 64.25±3.03 | 65.14±3.22 | 67.41±3.28 | **68.13±3.42** |
| UMIST | 62.56±2.73 | 68.74±3.16 | 69.46±3.11 | 70.37±3.52 | 72.03±2.30 | 70.90±3.25 | 69.27±2.46 | **72.32±2.42** | 72.03±2.56 |
| COIL | 76.34±2.06 | 77.85±1.69 | 80.21±2.16 | 82.21±2.31 | 83.51±2.42 | 80.57±2.47 | 81.16±2.63 | 83.68±1.95 | **84.18±2.21** |
| MSRA | 50.18±1.73 | 51.42±2.09 | 53.46±2.08 | 55.06±2.18 | 56.02±2.07 | 55.89±2.42 | 53.46±2.24 | 55.96±2.08 | **56.47±2.32** |

Table 6: ARI $(\%)$ of various methods and their standard deviations on data sets

| data sets | KKM | KFKM | KPKM | DEC | IDEC | DFKM | AE | AEL | LUOPSC |
|---|---|---|---|---|---|---|---|---|---|
| Dna | 33.28±1.49 | 34.61±3.06 | 36.41±2.12 | 38.14±1.68 | 39.47±3.51 | 37.32±1.73 | 36.48±1.62 | 39.53±2.04 | **40.15±3.05** |
| Pendigits | 60.52±2.43 | 63.28±1.88 | 61.26±1.76 | 65.33±2.49 | 66.74±0.83 | 65.40±1.77 | 65.32±0.79 | 65.58±1.41 | **66.23±1.50** |
| Iris | 71.56±1.46 | 72.89±2.36 | 73.42±3.55 | 75.68±2.49 | 76.29±2.87 | 77.31±2.57 | 74.42±2.13 | 78.21±2.25 | **79.42±1.89** |
| Satimage | 49.32±2.56 | 48.51±2.63 | 47.92±2.56 | 49.66±3.56 | 51.28±2.40 | 50.32±3.15 | 48.82±2.09 | 51.34±2.50 | **52.61±2.78** |
| Waveform | 28.34±2.77 | 30.29±2.42 | 31.24±2.09 | 36.73±2.29 | 38.68±1.42 | 35.28±2.08 | 36.67±1.09 | **38.80±2.08** | 37.43±2.19 |
| ORL | 19.57±2.26 | 20.29±2.31 | 22.36±3.09 | 22.48±1.52 | 22.95±1.62 | 21.32±2.41 | 22.31±1.87 | 22.42±3.5 | **23.01±1.62** |
| Yale | 20.53 ±1.46 | 21.49±3.42 | 20.54±3.81 | 21.59±4.06 | 23.26±3.88 | 24.29±3.73 | 21.72±3.83 | 24.49±3.61 | **25.81±3.37** |
| UMIST | 37.16±2.53 | 38.28±2.21 | 36.05±3.12 | 38.98±3.22 | 39.07±3.12 | 36.17±3.38 | 36.53±2.91 | 38.42±4.05 | **39.42±3.16** |
| COIL | 69.14±1.56 | 71.24±1.89 | 73.14±2.63 | 76.35±2.67 | 77.82±2.34 | 75.23±2.06 | 72.26±3.05 | **77.87±2.19** | 74.33±2.07 |
| MSRA | 48.56±2.37 | 50.54±2.08 | 56.38±2.15 | 57.28±2.28 | 57.89±2.06 | 56.41±2.51 | 56.42±2.06 | 57.95±2.74 | **58.40±2.62** |

Table 7: clustering accuracy(ACC) and ARI of various methods on two large-scale data sets

| data sets | KKM | KFKM | KPKM | DEC | IDEC | DFKM | AE | AEL | LUOPSC |
|---|---|---|---|---|---|---|---|---|---|
| MNIST(ACC) | 77.62±1.03 | 78.38±1.44 | 72.26±1.40 | 83.12±1.45 | 83.40±1.79 | 82.59±1.75 | 82.29±1.82 | 85.92±1.34 | **86.22±1.83** |
| MNIST(ARI | 65.13±2.06 | 66.26±2.31 | 67.40±1.92 | 73.32±2.07 | 74.91±2.52 | 72.17±1.26 | 68.86±0.97 | 77.42±1.50 | **78.31±1.47** |
| Fashion(ACC) | 53.11±2.07 | 56.75±2.82 | 54.32±3.07 | 55.91±2.08 | 55.62±2.48 | 53.29±2.51 | 62.32±2.14 | 73.21±2.29 | **77.32±2.36** |
| Fashion(ARI) | 40.53±2.17 | 41.28±2.76 | 39.87±2.52 | 41.96±2.53 | 42.28±3.02 | 40.52±3.32 | 40.55±2.73 | 42.39±2.62 | **44.54±2.04** |

Table 8: The running time of various methods on two large-scale data sets

| data sets | KDA | KDAL1 | RKFDA | DFDA | DWDA | AE | AEL | LUOP |
|---|---|---|---|---|---|---|---|---|
| MNIST(s) | 35 | 167 | 25 | 564 | 581 | 545 | 612 | 958 |
| Fashion(s) | 45 | 127 | 39 | 543 | 569 | 579 | 632 | 903 |

For the large-scale FashionMNIST data set, we carry out the experiment to check the effect of reduced dimensions and the number of neighbors. Figure 3 (a) denotes the correct rates of LUOP with the change of reduced dimensions, and Figure 3 (b) shows the correct rate of LUOP as the number of neighbors varies. From Figure 3 (a), we observe that the reduced dimensions affect the performance of LUOP. But when the dimensions of the samples exceed 10, our model can achieve good performance. From Figure 3 (b), we can see that it is not necessary to employ too many neighbors to obtain good better performance since we consider the samples from the same class. In addition, we visualize 2000 samples in a two-dimensional space via t-SNE. Figure 4 shows the experimental results. Figure 4 (a) denotes the visualization of original images via t-SNE, and Figure 4 (b-d) denote the results of LUOP in the case of different iterations. As can be seen from Figure 4, the embedding features in a two-dimensional space from LUOP are well separated.

We also discuss the effect of different regularization terms. Table 9 lists experimental results on the MNIST data sets. Since we consider the same encoder and decoder, $\lambda_4 = 0$ in our model, which means we explore self optimal transport in the same encoded space. When $\lambda_1 = 0$, we set $p_{j|i} = 1/k$, we optimize $p_i$ and parameters of autoencoders. When $\lambda_2 = 0$, we set $p_i = 1/n$, we optimize $p_{j|i}$ and parameters of autoencoders. When $\lambda_3 = 0$, we optimize $p_{j|i}$, $p_i$ and parameters of encoders since we do not consider decoders. In such a case we use the orthogonal technique

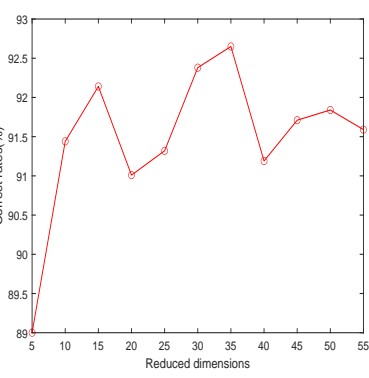
(a) Correct rates versus reduced dimensions

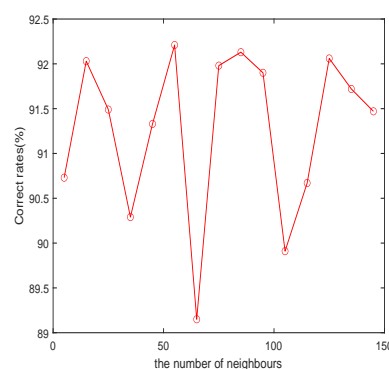
(b) Correct rates versus the number of neighbors

Figure 3: Performance of our model on the FashionMNIST data set

(torch.nn.utils.parametrizations.orthogonal)to avoid the degenerate solution. In addition, we explore the case that $q_{j|i} = 1/k$, i.e. $q_{j|i}$ adopts the uniform distribution instead of student $t$ distributions.

Table 9: The effect of different regularization terms

| Regularization parameters | | | Accuracy and standard deviations (%) |
|---|---|---|---|
| $\lambda_1 = 0$ | $\lambda_2 = 1000$ | $\lambda_3 = 0.1$ | $9.31 \pm 1.26$ |
| $\lambda_1 = 100$ | $\lambda_2 = 0$ | $\lambda_3 = 0.1$ | $10.46 \pm 2.41$ |
| $\lambda_1 = 100$ | $\lambda_2 = 1000$ | $\lambda_3 = 0$ | $15.76 \pm 2.46$ |
| $\lambda_1 = 100(q_{j|i} = 1/k)$ | $\lambda_2 = 1000$ | $\lambda_3 = 0.1$ | $9.06 \pm 2.55$ |

From Table 9, we observe that removing the reconstruction error results in the biggest performance loss since the reconstruction error preserves the global structure of data. Optimizing $p_{j|i}$ in our model can give better performance than fixing parameter $p_{j|i}$ in our model. It is beneficial to learn $p_i$ instead of fixing $p_i$. These experiments show that adding these regularization terms can improve the performance of our model. For large-scale data sets, we test a simple fully connected network with a three-layer encoder and a three-layer decoder on the above experiments. In fact, other neural networks can also be used in our model. Here we test Resnet50 structure on large-scale data sets. Note that we compare our model with the Resent50 structure to other supervised learning such as supervised contrastive learning(SupCon)(Khosla et al., 2020) and variational supervised contrastive learning(VarCon)(Wang et al., 2025). Unlike some methods, we do not consider any data augmentation technique. In addition, we add the label noise to the training set. Specifically, we randomly change the label of samples in the training set. Table 10 lists the experimental results on large-scale data sets.

Table 10: Classification error rates on two large-scale data sets

| Data Sets | SupCon | VarCon | LUOP |
|---|---|---|---|
| MNIST | 3.45 | 3.12 | 3.98 |
| Fashion | 7.85 | 6.89 | 7.01 |
| MNIST with label noise(20%) | 6.31 | 6.05 | 5.21 |
| Fashion with label noise(20%) | 9.22 | 8.36 | 8.07 |

From Table 10, we observe that our model can obtain competitive performance with SupCon and VarCon when more complex neural networks are explored. It is noted that our model can obtain better performance than other model since using optimized weights can suppress the label noise in training set.

There are several hyperparameters in the proposed model since these parameters determine trade-offs in several terms. We let $\lambda_2=1000$ so that $\{p_i|i=1,\cdots,n\}$ approach uniform distributions. If we consider that $y_1,\cdots,y_k$ are taken from $x_1,\cdots,x_n$, then we set $\lambda_4 = 0$. In such a case, we first

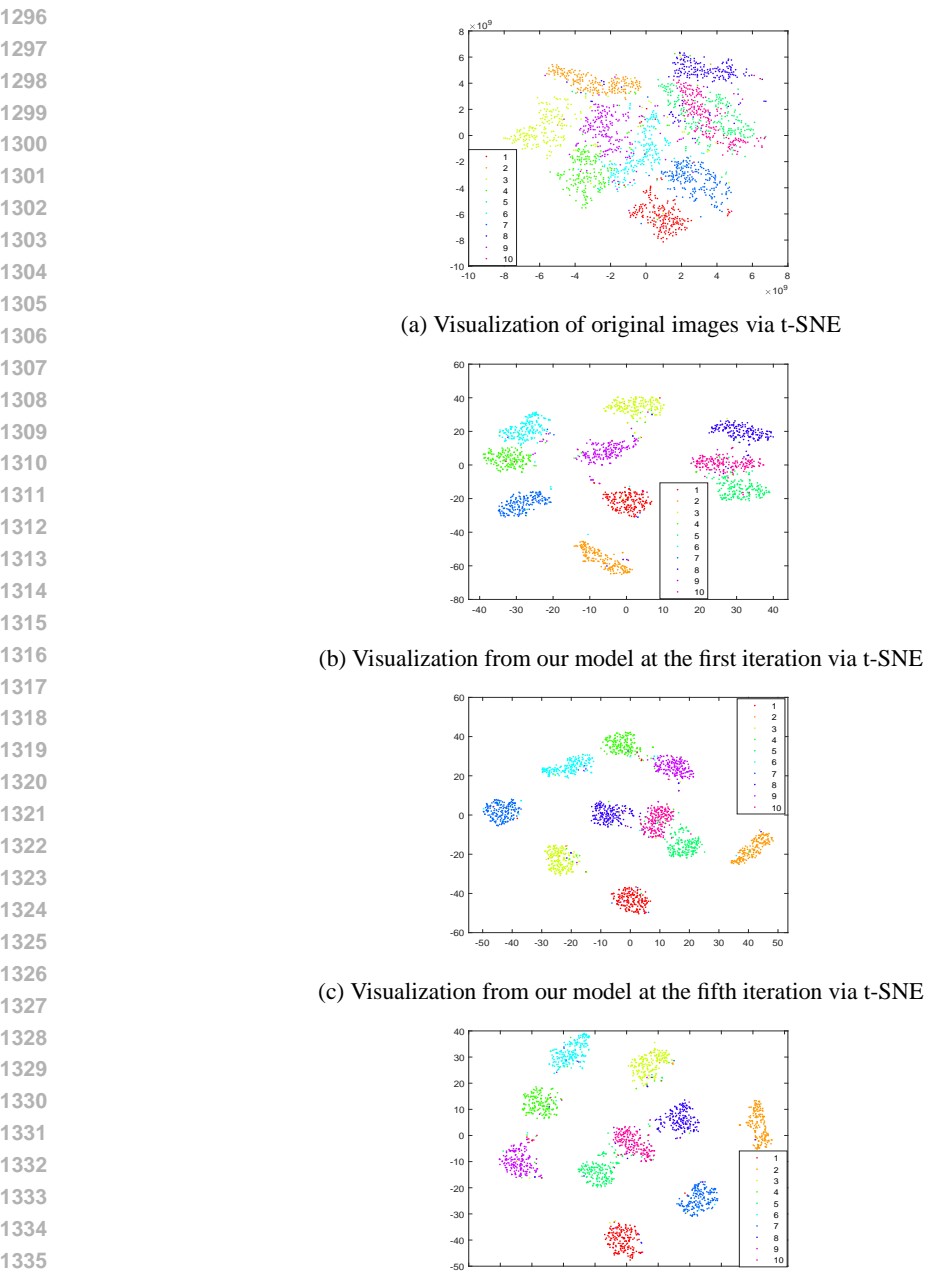

(a) Visualization of original images via t-SNE

(b) Visualization from our model at the first iteration via t-SNE

(c) Visualization from our model at the fifth iteration via t-SNE

(d) Visualization from our model at the tenth iteration via t-SNE

Figure 4: Visualization of 2000 images on the FashionMNIST data set

explore the effect of different $\lambda_1$ and $\lambda_3$ on supervised learning tasks. To this end, we randomly choose half of samples from each person to form the training set and others are used for testing on the ORL data set. Assume that the reduced dimension is equal to the number of classes (40) and the hyperparameters $\lambda_1$ and $\lambda_3$ take values from $\{0.001, 0.01, 0.1, 1, 10, 100, 1000\}$. Thus each parameter takes seven values. We also report the experimental results over ten runs. Figure 5 shows the experimental results on the classification problem, where the $x$-axis denotes the hyperparameter $\lambda_3$, the $y$-axis denotes the hyperparameter $\lambda_1$, and the $z$-axis denotes the classification error rates of our model.

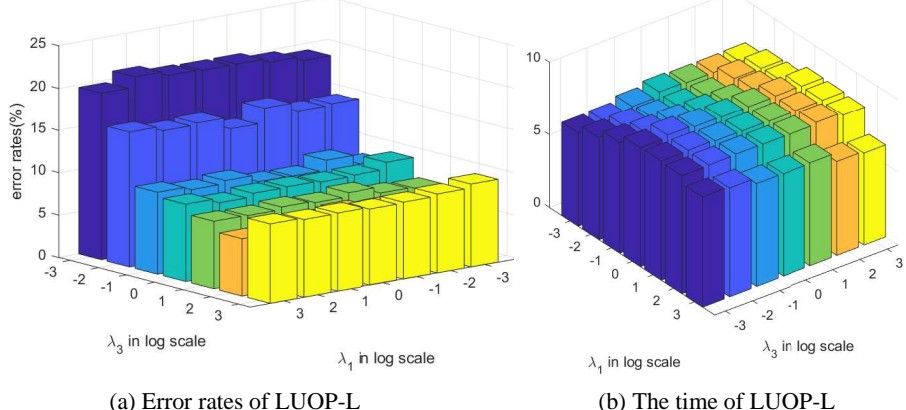

(a) Error rates of LUOP-L

(b) The time of LUOP-L

Figure 5: Performance of LUOP-L with varying hyperparameters

As can be seen from Figure 5, the error rates of the proposed model vary with the change of hyperparameters. It is found that the error rates of our model are very high when the hyperparameter $\lambda_3$ takes relatively small values. We observe that $\lambda_3$ is more sensitive than $\lambda_1$ in affecting the performance of the model. From Figure 5, we see that the running time of our model is affected by the hyperparameters. Figure 6 shows the experimental results on the clustering problem. From Figure 6, we find that the hyperparameters affect the performance of LUOPSC in the clustering problem. Overall, the experiments indicate that we need to select proper parameters to attain the best performance in real applications. In fact, the cross-validation is often employed to select optimal parameters.

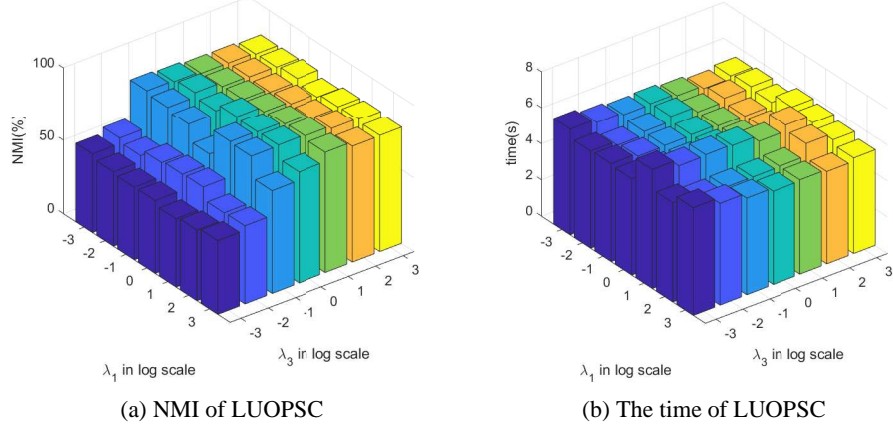

(a) NMI of LUOPSC

(b) The time of LUOPSC

Figure 6: Performance of LUOPSC with varying hyperparameters

