# OpenReview forum: "Local and Unbalanced Optimal Transport for Feature Learning with Probabilistic Guarantees"
_ICLR.cc/2026/Conference — Submitted to ICLR 2026_

### Official Review · Reviewer_QDkJ · 2025-10-29

**Soundness:** 2
**Presentation:** 2
**Contribution:** 2
**Rating:** 2
**Confidence:** 3

**Summary:**

This paper proposes a local and unbalanced optimal transport (OT) formulation for feature learning.
Specifically, the original unbalanced OT framework is modified with the conditional distribution to form a new formulation. Then regularization terms are added to avoid trivial solution.
As to the optimisation, alternative updating w.r.t. each set of parameters are designed for $p_{j|i}, p_i, $ and $\theta, \bar{\theta}, \phi, \bar{\phi}$. The first two sets are strong convex problems with closed-form solution, while the third one is nonconvex and be optimised with SGD.
Theoretical analysis gives the bound under the Rademacher complexity theory.

Experiments are conducted on supervised learning, clustering, and set-valued objects. Several baselins are compared, with the proposed LUOP and LUOP-L performing better.

**Strengths:**

## Strengths
- Conditional distribution is incorporated into the unbalanced optimal transport formulation to derive a new local and unbalanced OT framework. Alternative updating is proposed to solve the new problem.
- Rademacher complexity theory is adopted to derive two bounds for the proposed formulation.
- Experiments are conducted on supervised learning, clustering, and set-valued objects. With several baseline methods compared, the proposed achieves better performance.

**Weaknesses:**

## Weaknesses
- The main technical contribution is the introduction of conditional distributions, which transforms Eq. (1) to Eq. (2), and removing the third term. From the OT perspective, this contribution is incremental based on unbalanced OT framework.
Regularization terms in Eq. (4) are just common practice in feature learning, and cannot be regarded as new contribution in the context of feature learning.
- The optimisation process is alternative optimising (coordinate descent), which is commonly-used as the feasible solution. It requires alternating between variables.
- Although supervised learning, clustering, and set-valued objects tasks are conducted in this paper, the baseline and comparison methods are relatively out-dated and not strong/relevant opponents to the proposed LUOP method.
> All baselines are earlier than 2022.
> No unbalanced OT or local OT variants are included in the comparison or formulation discussion, e.g., [R1-R4], to name a few.
> No task-specific SOTA methods are compared in each section.
- Related work only appears in Appendix and is very short, lacking necessary local/unbalanced OT formulations and variants.

[R1] Thual, Alexis, et al. "Aligning individual brains with fused unbalanced Gromov Wasserstein." Advances in neural information processing systems 35 (2022): 21792-21804.

[R2] Chizat, Lenaic, et al. "Scaling algorithms for unbalanced optimal transport problems." Mathematics of computation 87.314 (2018): 2563-2609.

[R3] Séjourné, Thibault, François-Xavier Vialard, and Gabriel Peyré. "The unbalanced gromov wasserstein distance: Conic formulation and relaxation." Advances in Neural Information Processing Systems 34 (2021): 8766-8779.

[R4] Benamou, Jean-David, et al. "Iterative Bregman projections for regularized transportation problems." SIAM Journal on Scientific Computing 37.2 (2015): A1111-A1138.

**Questions:**

Are there any recent local/unbalanced OT framework most relevant to the proposed method?

Are there any SOTA methods for each of the task experimented, i.e., supervised learning, clustering, and set-valued objects?

Eq. (1) has two KL regularization terms, while the proposed Eq. (2) only has one. What is the consideration and intuition?

---

> ### Author Response · Authors · 2025-11-20
>
> Dear Reviewer QDkJ. We highly appreciate your time and effort to review our submission. We have uploaded a new version for our paper.
>
> 1) The main technical contribution is the introduction of conditional distributions, which transforms Eq. (1) to Eq. (2), and removing the third term. From the OT perspective, this contribution is incremental based on unbalanced OT framework.
> Regularization terms in Eq.(4) are just common practice in feature learning, and cannot be regarded as new contribution in the context of feature learning.
>
> Answer: If we only decompose joint distributions into conditional distributions and marginal distributions. This indeed is an incremental job for optimal transport. However, we combine the autoencoder into optimal transport and perform the optimal transport in the encoded space.
> This is also one contribution of this work. In addition, we extend our model for clustering tasks and set-valued classification. Most importantly, we theoretically analyze the generalization bound from the viewpoint of the statistical learning theory. We think that this is  the first time to analyze the optimal transport from the viewpoint of the statistical learning theory.
>
>
>
>
> 2) The optimisation process is alternative optimising (coordinate descent), which is commonly-used as the feasible solution. It requires alternating between variables.
> Although supervised learning, clustering, and set-valued objects tasks are conducted in this paper, the baseline and comparison methods are relatively out-dated and not strong/relevant opponents to the proposed LUOP method.
> All baselines are earlier than 2022.
>
> Our model can use linear encoders and decoders. Thus in such a case,  our model can be used to deal with small data sets. So we can compare our methods with some classical feature learning methods. It is an interesting result that deep learning techniques cannot obtain good results on small data sets. In fact, we can employ any autoencoder to our models so that our model can be used to deal with large-scale data sets. We also make experiments in terms of Resnet networks(Table 10). It is found that our model can obtain competitive performance in terms of SOTA.
>
>
> 3)  No unbalanced OT or local OT variants are included in the comparison or formulation discussion, e.g., [R1-R4], to name a few.
> No task-specific SOTA methods are compared in each section.
> Related work only appears in Appendix and is very short, lacking necessary local/unbalanced OT formulations and variants.
>
>
> These variants of optimal transport are mainly used to transport data in different domains, and they perform optimal transport in the original space. Our model mainly uses optimal transport and autoencoders to perform optimal transport in  the encoded space. In the new version, we cite these papers and point out the difference between our model and these variants.
>
>
> Questions:
> 4) Are there any recent local/unbalanced OT framework most relevant to the proposed method?
>
> Nowadays, there are numerous variants of optimal transport. Some works use neural networks to learn  joint distributions and continuous  conditional distributions. Our work mainly lies in learning the discrete conditional distributions in the encoded space.  We think that jointly learning the conditional distributions and autoencoders is one of the main contributions of our work.
>
>
> 5) Are there any SOTA methods for each of the task experimented, i.e., supervised learning, clustering, and set-valued objects?
>
>
>  Answer: On small data sets, our method can achieve better performance than other methods. On large-scale data sets, using our model with a simple encoder and decoder gives better performance. In these large-scale data sets, our model does not give the SOTA. In the new version,
> we change our encoders and use Resnet networks. and I find that our model can achieve competitive performance when we fix the hyperparameters of our methods. However, when the training set contains the label noise, our model can obtain better performance than other models since we learn the conditional distributions.
>
> 6) Eq. (1) has two KL regularization terms, while the proposed Eq. (2) only has one. What is the consideration and intuition?
>
> Answer:  For unbalanced optimal transport, its two marginal distributions are relaxed. When we fix one marginal distribution for Dirac measure in unbalanced optimal transport. Thus we obtain Eq.(2). Hence Eq.(2) is a special case of unbalanced optimal transport. Note that Eq.(2) only consider a neighborhood of a specific  data point. Thus Eq.(2) only  uses a regularization term. This explains why we use conditional distributions instead of joint distributions.  In fact, Eq.(2) is a part of our objective function.

---

> > ### Comment · Reviewer_QDkJ · 2025-11-26
> >
> > Thank the authors for the detailed reply and rebuttal.
> >
> > After carefully read the rebuttal, some of my questions are partly addressed, but my main concerns and scores remain.
> > 1. "perform the optimal transport in the encoded space" can slightly increase the novelty but overall the formulation and problem itself are still incremental in the OT area.
> > 2. "our model can be used to deal with small data sets". Deal with small data sets only is not realistic and lacks future application potential.
> > "deep learning techniques cannot obtain good results on small data sets". This might be due to the weak baseline methods compared (before 2022).
> > 3. Some recent papers are cited, but it still lacks detailed comparison.
> > 4. This point is not well addressed.
> > 5. "In these large-scale data sets, our model does not give the SOTA."
> > 6. My question is answered.
> >
> > Therefore, I will keep my score.

---

### Official Review · Reviewer_rJxC · 2025-10-29

**Soundness:** 2
**Presentation:** 1
**Contribution:** 2
**Rating:** 2
**Confidence:** 3

**Summary:**

The paper proposes a feature-learning approach based on conditional and unbalanced optimal transport. It introduces conditional distributions in terms of the Kullback-Leibler divergence, claims closed-form updates under block coordinate descent, and adds anchors and an autoencoder to learn an embedding for clustering and classification of set-valued objects. Generalization bounds via Rademacher complexity are provided.

**Strengths:**

The approach is evaluated on diverse tasks and datasets.

**Weaknesses:**

- The problem definition of feature learning is unclear. The paper uses “feature learning” interchangeably with representation learning, dimensionality reduction, metric learning, and clustering.
- Section 2.2 is overcomplicated and hard-to-follow formulation. It is lengthy, mixes motivations with derivations, and introduces Eq. (4) in a way that seems unnecessarily complex for the same-source setting.
- The paper reads like a technical report without providing the motivation of why each component is considered plus the overcomplicated notation does not help. The paper explains what is added (conditionals, anchors, autoencoder) but not why each piece is needed and which phenomenon it addresses.
- Theorems 1–2 are presented without explaining why they are interesting or what the bounds imply practically. The proof in the in appendix is hard to follow. What is the interpretation? Could you provide a pointer to the lines in the proof where key ideas occur? Is there a sanity-check experiment showing the bound’s qualitative predictions?
- There are many proposed components in the model, but there is no ablation study. Unclear how each component contributes to the final performance
- Citing very recent or narrow works as the first mention of core notions (OT, unbalanced OT, KL, f-divergence) is misleading:
    - Intro/lines 83–84, 117: Use standard OT and UOT references when first introducing them; recent applied papers can be cited later for variants.
    - Line 121 (KL) and line 157 (f-divergence): Cite canonical sources for definitions, not recent downstream applications.
- Considering that various neighborhoods are explored in the graph community, but there is no comparison or reference to them.
- The hyperlinks to references, figures, and tables are broken
- The autoencoder appears with no reference or reason for why latent-space OT is preferable to data-space OT here.
- The autoencoder is first mentioned with no reference
- “Alternating optimization” is listed as a contribution, but this is a standard technique. Perhaps the authors could clarify what is novel (a provably convergent block structure? closed-form updates?).
- Lines 213–215 (Eq. 5): The strong convexity claim and closed-form derivation need either an inline proof sketch or an explicit pointer to a lemma. Lines 222–227 (Eq. 6): Same issue.
- In lines 252-253: I do not see how p_{j|i} or p_i is defined in subsection 3.2
- It is not clear what the key differences are between the proposed method and existing work, eg, Wasserstein Discriminant Analysis or kernel-based Discriminant Analysis.
- in line 27: missing a period in the sentence “.. to classify set-valued objects”
- Notation abuse: X and Y were used as spaces in Section 1, but then later were defined as random vectors in Section 2.1
- In Eq. (1), p_{ij}, U(a,b),p and pe_k  are not defined
- In line 138, the Wasserstein space is not defined
- in line 141, the Wasserstein distance  $\\bar{W}\_i^{\\bar{r}}$ is minimized over a scaler p_{j,i}?
- Why Eq.(2) is considered for the use of a special case of the unbalanced optimal transport for effectively learn p_{j|i} in the conditional distribution
- The used notations are very complex and hard to follow. Perhaps the authors could include a table of all the notation.
- The appendix is referred to without a pointer - hard to navigate what to look for and point to in the relevant context

**Questions:**

See above

---

> ### Author Response · Authors · 2025-11-20
>
> Dear Reviewer  rJxC
>
>  We highly appreciate your time and effort to review our submission. We have uploaded a new version for our paper.
>
> 1) The problem definition of feature learning is unclear. The paper uses “feature learning” interchangeably with representation learning, dimensionality reduction, metric learning, and clustering.
>
>  Answer: In the new version, we try to avoid causing confusion with these notations. In fact, feature learning aims to learn good feature representations of data so that using these features can effectively perform classification and clustering tasks.
>   Representation learning originates from deep learning. By using self supervised learning, one can achieve good feature representations to contribute to downstream tasks such as classification and clustering problems.
>   Dimension reduction is to reduce the dimension of samples so that samples can be understood in a low-dimensional space. Using metric learning  can find good feature representations of data. By combining clustering into autoencoders,  one can obtain good feature representations of samples.   Overall,  these techniques help us find good feature representations of samples.
>
>
>
> 2) Section 2.2 is overcomplicated and hard-to-follow formulation. It is lengthy, mixes motivations with derivations, and introduces Eq. (4) in a way that seems unnecessarily complex for the same-source setting.
>
> Answer: To help readers  understand our objective function, we have to start by considering conditional distributions (local information of samples). From the viewpoint of  the optimal transport, the transport occurs in the same space. In the case of autodercoders, we can explore that samples in the encoded space can be transported. If different sources are adopted, we have to use different encoders to encode samples so that they can be transported in the encoded space. We reserve the generic notations.
>
>
>
> 3) The paper reads like a technical report without providing the motivation of why each component is considered plus the overcomplicated notation does not help. The paper explains what is added (conditionals, anchors, autoencoder) but not why each piece is needed and which phenomenon it addresses.
>
> Answer: Some optimal transport methods are explored in the original space. In this paper, we try to explore transport samples in the encoded space by autoencoders. Conditional information can be used to characterize the local information of samples.  Anchors are adopted from the kernel-based learning, where large-scale kernels can be reduced. We use anchors to generalize our model to be suitable for unsupervised learning, which is also similar to the clustering methods.
>
>
> 4) Theorems 1–2 are presented without explaining why they are interesting or what the bounds imply practically. The proof in the in appendix is hard to follow. What is the interpretation? Could you provide a pointer to the lines in the proof where key ideas occur? Is there a sanity-check experiment showing the bound’s qualitative predictions?
>
> Answer: These theorems are inspired by studying the generalization bound of k-means. From the viewpoint of the statistical learning theory, each learning algorithm has the corresponding generalization bounds. Nowadays, only limited neural networks have derived generalization bounds. Moreover, the generalization  bounds are loose for neural networks. These generalization bounds only show that optimizing these generalization bounds may obtain good generalization performance. So our generalization bounds depend on Rademacher complexities of neural networks. Estimating of Rademacher complexities is challenging for neural networks even if simple networks are adopted. Our generalization bounds depend on  Rademacher complexities of neural networks, where we use known results from neural networks.    Therefore, we cannot use toy examples to show these generalization bounds.
>
>
>
> 5) There are many proposed components in the model, but there is no ablation study. Unclear how each component contributes to the final performance.
>
> Answer: In the new version, we make some experiments to show that some components are useful.
>
>
> 6) Citing very recent or narrow works as the first mention of core notions (OT, unbalanced OT, KL, f-divergence) is misleading:
> Intro/lines 83–84, 117: Use standard OT and UOT references when first introducing them; recent applied papers can be cited later for variants.
>
> Line 121 (KL) and line 157 (f-divergence): Cite canonical sources for definitions, not recent downstream applications.
>
> Answer: We modify them.

---

> ### Author Response · Authors · 2025-11-20
>
> 7) Considering that various neighborhoods are explored in the graph community, but there is no comparison or reference to them.
>
> Answer: Although various neighborhoods are explored in the graph community, our aim is to explore feature learning in encoded spaces by using autoencoders.  We  compare our work with  some kernel-based methods and methods based on autoencoders. It has been shown that  kernel-based methods can capture the intrinsic structure than linear graph feature learning methods.  Currently, graph neural networks can be used to learn the data with graph structures, but we do not explore these types of neural networks since our generalization  bound is not suitable for graph neural networks. Therefore, we do not focus on the graph structure based on local information.
>
>
> 8) The hyperlinks to references, figures, and tables are broken
>
> Answer: We correct these points.
>
> 9) The autoencoder appears with no reference or reason for why latent-space OT is preferable to data-space OT here.
>
>  Answer: We think that autoencoders may be simple notations for our paper, but we give some references in the our new version.  For high-dimensional data, we explored the encoded space by using optimal transports so that good feature representations can be found in the encoded space. Thus good feature representations are beneficial for classification and clustering tasks.
>
> 10) The autoencoder is first mentioned with no reference.
>
>  Answer: We mention autoencoders when they appear for the first time.
>
> 11) “Alternating optimization” is listed as a contribution, but this is a standard technique. Perhaps the authors could clarify what is novel (a provably convergent block structure? closed-form updates?).
>
> Answer: Alternating optimization is a standard optimization technique. In fact, we can optimize our model based on other optimization algorithms. But adopting block structures can give the closed-form of conditional distributions, which is an interesting result for our method in the case of  alternating optimization,  which may make our method different from the sinkhorn method in the optimal transport.
>
> 12) Lines 213–215 (Eq. 5): The strong convexity claim and closed-form derivation need either an inline proof sketch or an explicit pointer to a lemma. Lines 222–227 (Eq. 6): Same issue.
>
> Answer: In the new version, the strong convexity needs an additional conditionals(Melbourne, James,Strongly Convex Divergences,Entropy, 22, 2020). The convexity of our objective comes from the convexity of the negative entropy function. We give a simple derivation of one step in our algorithm in the appendix. In the new version, we reserve the convexity to allow for the zero probability.
>
> 13) In lines 252-253: I do not see how $p_{j|i}$ or $p_i$ is defined in subsection 3.2.
>
> Answer: These definitions can be found in subsection 2.3 instead of subsection 3.2 in the old version. In the new version,  these definitions can be found in subsection 3.3.
> $p_{j|i}$ has the closed-form  solution defined in subsection 3.3(a) or $p_i$ has the closed-form solution defined in subsection 3.3 (b).
>
> 14) It is not clear what the key differences are between the proposed method and existing work, eg, Wasserstein Discriminant Analysis or kernel-based Discriminant Analysis.
>
> Answer: Wassersstein discriminant analysis  adopts joint distributions and does not explore conditional distributions. Wasserstein discriminant analysis does not explore  the encoded space, but our model explores the encoded space.
> Kernel-based discriminant analysis employs  kernel functions to perform feature learning. In large-scale kernels, anchors are used to reduce the computational complexity of the kernel function. They are derived from the discriminant criteria. Our model is derived from the optimal transport plus some regularization terms, and our model explores the encoded space from autoencoders.
>
>
>
>
> 15) in line 27: missing a period in the sentence “.. to classify set-valued objects”
>
> Answer: We correct this point.
>
>
> 16) Notation abuse: X and Y were used as spaces in Section 1, but then later were defined as random vectors in Section 2.1.
> We change these notations to avoid confusion.
>
> In Eq. (1), $p_{ij}$, $U(a,b)$,p and $pe_k$ are not defined
>
>  Answer: In the new version, we give clear definitions.
>
> 17) In line 138, the Wasserstein space is not defined.
>
>  Answer: We cite one book for this definition since it is not straightforward.
>
>
> 18) in line 141, the Wasserstein distance
>  is minimized over a scaler $p_{j,i}$?
>
>  Answer: This is correct if  autoencoers are fixed.

---

> > ### Author Response · Authors · 2025-11-20
> >
> > 19) Why Eq.(2) is considered for the use of a special case of the unbalanced optimal transport for effectively learn $p_{j|i}$ in the conditional distribution
> >
> >  Answer: For unbalanced optimal transport, its two marginal distributions are relaxed. When we fix one marginal distribution for Dirac measure in unbalanced optimal transport. Thus we obtain Eq.(2). So Eq.(2) is a special case of unbalanced optimal transport.
> >
> > 20) The used notations are very complex and hard to follow. Perhaps the authors could include a table of all the notation.
> >
> >  Answer In the new version, we list a table for these notations.
> >
> > 21) The appendix is referred to without a pointer - hard to navigate what to look for and point to in the relevant context
> >
> >  Answer: We use  pointers for appendices.

---

### Official Review · Reviewer_xXdB · 2025-10-30

**Soundness:** 3
**Presentation:** 3
**Contribution:** 3
**Rating:** 6
**Confidence:** 4

**Summary:**

This paper introduces a framework for feature learning that leverages Local and Unbalanced Optimal Transport (OT) within an autoencoder architecture. The core innovation is replacing the standard joint probability measure of Optimal Transport with conditional distributions. For each data point $x_i$, a conditional measure $p\left(y \mid x_i\right)$ is constructed on a set of its local/relevant neighbors $\boldsymbol{y}_{j \mid i}$ in the target space.


The objective function is a generalized loss that combines: An unbalanced OT cost between the Dirac measure $\delta_{f_\theta\left(x_i\right)}$ and the conditional measure $p\left(g_\phi(y) \mid x_i\right)$ in the embedding space and Reconstruction errors for the autoencoders, to prevent trivial solutions.


The model is optimized via an alternating optimization technique (Block Coordinate Descent) with closed-form solutions for the conditional ( $p_{j \mid i}$ ) and marginal ( $p_i$ ) probability vectors. Furthermore, the paper provides a generalization bound for the model based on Rademacher complexity. The framework is empirically validated on classification and clustering tasks, and is extended to a classifier for set-valued objects.

**Strengths:**

The paper's central claims are well-supported. The theoretical formulation introduces an objective function that balances OT cost, KL-divergence regularization, and autoencoder reconstruction. The derivation of a generalization bound via Rademacher complexity (Theorem 1) further strengthens the theoretical foundation by offering probabilistic guarantees for the feature learning process. The experimental setup is comprehensive, covering standard classification and clustering, as well as an application to set-valued object classification.


The paper is generally well-written and clear. The problem formulation is logically built, starting from preliminaries on Optimal Transport and then introducing the local, conditional, and unbalanced components. The figures and equations are helpful.


This work makes a good contribution to the field of feature learning via Optimal Transport.


The resulting loss formulation is a novel combination of concepts from autoencoders and Optimal Transport. The model's extension to classifying set-valued objects (LUOPC) suggests a broader applicability for the proposed local optimal transport framework beyond standard feature learning tasks.

- The algorithm is well-derived, with the outcome being the closed-form, analytic solutions for the conditional and marginal distributions ( $p_{{j} {i}}, {p}_{{i}}$ ) during the alternating optimization. Furthermore, the provision of a generalization bound via Rademacher complexity anchors the method in rigorous theoretical analysis.


- The framework is generic and successfully applied to multiple learning tasks (classification, clustering) and a new problem (set-valued classification), showing its practical utility and broad potential in feature representation.

**Weaknesses:**

- The primary weakness of this paper lies in the lack of a comprehensive literature review on works that use conditional optimal transport, despite the paper's main contribution being the description of local information of data points via optimal transport. For instance:
[1] Manupriya, Piyushi, et al. "Consistent optimal transport with empirical conditional measures." International Conference on Artificial Intelligence and Statistics. PMLR, 2024.
Additionally, since the authors primarily use unbalanced optimal transport (OT), a discussion of relevant literature on this approach would have been beneficial.


- Some findings are trivial, such as in lines 252-254, where it is quite evident that by increasing the weight $\lambda$ of the $\lambda K L(p \| q)$ term to infinity and minimizing with respect to $p$, it forces $p$ to equal $q$.


- In formulation (1), which appears to be in the unbalanced OT form, when applying the KL penalty to the two marginals, the constraint $p_{i j} \in U(a, b)$ should be relaxed. If not, the penalty becomes nonsensical.


- The formulation includes several hyperparameters $\left(\lambda_{1 \rightarrow 4}\right)$, and the paper should include a discussion on the ablation study regarding their effects.


- The authors should elaborate more on the convergence of Algorithm 1, particularly in terms of the block coordinate descent method, rather than simply mentioning it in a single sentence. There are many examples where applying block coordinate descent does not converge to a stationary point, which warrants further clarification.

**Questions:**

Beside some **concerns in the Weakness that can be taken as relevant questions well**, there is one more question for the authors about the choice of prior $q_{j \mid i}:$ In the experiments, the paper uses a long-tailed student t distribution for the prior $\boldsymbol{q}_{j \mid i}$. Could the authors justify this specific choice and explore the impact of other priors, such as a uniform distribution or a simpler proximity-based kernel, on the final feature quality and performance?

---

> ### Author Response · Authors · 2025-11-20
>
> Dear Reviewer xXdB
>
> We highly appreciate your time and effort to review our submission. We have uploaded a new version for our paper.
>
> 1) The primary weakness of this paper lies in the lack of a comprehensive literature review on works that use conditional optimal transport, despite the paper's main contribution being the description of local information of data points via optimal transport. For instance: [1] Manupriya, Piyushi, et al. "Consistent optimal transport with empirical conditional measures." International Conference on Artificial Intelligence and Statistics. PMLR, 2024. Additionally, since the authors primarily use unbalanced optimal transport (OT), a discussion of relevant literature on this approach would have been beneficial.
>
>  Answer: We cite some related work in the new version. In  consistent optimal transport, neural networks are used to model conditional measure. Our model jointly learns discrete conditional distributions and encoded space.
>
> 2) Some findings are trivial, such as in lines 252-254, where it is quite evident that by increasing the weight.
>
> Answer: However, we point out that these findings hold true by using the closed-form of conditional probability.
>
> 3) In formulation (1), which appears to be in the unbalanced OT form, when applying the KL penalty to the two marginals, the constraint
>  should be relaxed. If not, the penalty becomes nonsensical.
>
> Answer: We do not correctly describe unbalanced optimal transport.  In formulation (1),  the joint distribution has relaxed marginal distributions instead of $a$ and $b$. We correct them in the new version.
>
>
> 4) The formulation includes several hyperparameters
> , and the paper should include a discussion on the ablation study regarding their effects.
>
> Answer: In the new version, we make experiments for these components.
>
>
> 5) The authors should elaborate more on the convergence of Algorithm 1, particularly in terms of the block coordinate descent method, rather than simply mentioning it in a single sentence. There are many examples where applying block coordinate descent does not converge to a stationary point, which warrants further clarification.
>
>
> Answer: This may be one disadvantage  of our method due to the use of the neural networks.  In classical optimization theory, some works have discussed the convergence of  alternating optimization.(  Convergence of alternating optimization (Bezdek, James C.  and  Hathaway, Richard J, Neural, Parallel and Scientific Computations). However, due to the use of deep neural networks, proving its convergence of our method may be challenging. Hence we remove the statement of the convergence of our model. We find that the trend of the value of the loss function decreases when  we directly use optimization tools.
>
>
> 6) Beside some concerns in the Weakness that can be taken as relevant questions well, there is one more question for the authors about the choice of prior
>  In the experiments, the paper uses a long-tailed student t distribution for the prior
> . Could the authors justify this specific choice and explore the impact of other priors, such as a uniform distribution or a simpler proximity-based kernel, on the final feature quality and performance?
>
>
> Answer: In fact, any prior distribution can be employed in our model. We use a long-tailed student t distribution. This is inspired from  the deep clustering model via the t distribution (DEC) (Xie et al., 2016). In the new version, we perform the experiments for the uniform distribution(See Table 9 in the appendix). We find that the uniform distribution cannot give better performance since it gives the same weights for samples in the neighborhood.

---

### Official Review · Reviewer_xnAA · 2025-10-31

**Soundness:** 3
**Presentation:** 2
**Contribution:** 2
**Rating:** 4
**Confidence:** 3

**Summary:**

The paper describes a feature learning framework that combines local and unbalanced optimal transport  within an autoencoder architecture.

The approach replaces global joint distributions with conditional measures $p(y \mid x_i)$ defined on local neighborhoods $\mathbf{y}_{j \mid i}$ to model data relationships.

Its objective function integrates an unbalanced OT cost between $\delta_{f_\theta(x_i)}$ and $p(g_\phi(y) \mid x_i)$ in the latent space together with reconstruction terms from the autoencoders.

The model is optimized by block coordinate descent with closed-form updates for both conditional and marginal probabilities.

A generalization bound based on Rademacher complexity is derived, and experiments are reported on classification, clustering, and set-valued data.

**Strengths:**

The paper formulates a feature learning framework that integrates local and unbalanced optimal transport (OT) with autoencoders.
The objective is expressed as a combination of an unbalanced OT term
$L_{OT}(\delta_{f_\theta(x_i)}, p(g_\phi(y)\mid x_i))$,
a KL-divergence regularizer, and a reconstruction loss for the autoencoder components.
A conditional measure $p(y\mid x_i)$ is defined over the neighborhood $\mathbf{y}_{j\mid i}$ for each sample $x_i$,
replacing the global joint OT formulation with a local, sample-wise mapping.

During optimization, the variables $p_{j\mid i}$ and $p_i$ are updated in closed form under an alternating procedure.

Theoretical analysis introduces a generalization bound based on Rademacher complexity to describe the expected loss behavior.

**Weaknesses:**

Objective/constraints may be inconsistent:
Eq.(1) includes both a hard marginal constraint $p_{ij}\in U(a,b)$ and KL penalties on the marginals:
$
L_{OT} = \sum_{i,j} \rho(x_i,y_j)^{\bar r} p_{ij} + \lambda_1 \mathrm{KL}(p e_k \| a)+\lambda_2 \mathrm{KL}(p^\top e_n \| b),
$
see lines with “$p_{ij}\in U(a,b)$” and the two KL terms. If KL terms are used to relax marginals, the hard set $U(a,b)$ should be relaxed as well; otherwise the role of the penalties is unclear.

Role of priors $q_{j\mid i}, q_i$ is under-specified empirically:
The paper sets $q_{j\mid i}$ via a Student-$t$ kernel in the original space and discusses $q_i$, but their effects are not isolated.

Set-valued classification needs stronger baselines and clearer protocol.
When set size is 1 and $\lambda_1\to\infty$, the model degenerates toward a NN-style rule; comparisons to strong NN/prototypical baselines and train/test details are limited.

No ablations for key hyperparameters:
The objective exposes multiple hyperparameters $(\lambda_{1},\ldots,\lambda_{4})$ and design choices (e.g., neighborhood size $k$), but their effects are not isolated.

Scalability reporting is missing:
Although per-step complexities are stated, there is no wall-clock or memory usage across dataset sizes.

**Questions:**

1. In Eq.(1), are marginals enforced hard via $p_{ij}\in U(a,b)$ or soft via KL on marginals? If both are used, what is the KL term’s role? Please run an ablation: (a) hard-only (keep $U(a,b)$; no KL), (b) soft-only (drop $U(a,b)$; KL$>$0), (c) both.

2. Please include runtime and peak-memory tables on small/medium/large datasets vs. baselines with identical hyperparams.

3. How much performance comes from the priors $q_{j\mid i}, q_i$ vs. the transport term? Please report an ablation (i) with priors and (ii) $\lambda_1,\lambda_2\in\{0,\text{tuned}\}$, including accuracy/NMI and sensitivity curves.

4. Please add ablations for $\lambda_{1},\ldots,\lambda_{4}$, neighborhood size $k$, and the distance/order choice; report mean $\pm$ CI over multiple seeds and include sensitivity curves.

---

> ### Author Response · Authors · 2025-11-20
>
> Dear Reviewer xnAA
>
> We highly appreciate your time and effort to review our submission. We have uploaded a new version for our paper.
>
> 1) Objective/constraints may be inconsistent: Eq.(1) includes both a hard marginal constraint  and KL penalties on the marginal:
>  see lines with. If KL terms are used to relax marginal, the hard set
>  should be relaxed as well;
>
>  Answer: We do not correctly describe formulation (1) in the old version.  In fact, the joint distribution has relaxed marginal distributions instead of $a$ and $b$.  We correct them in the new version.
>   Thus, these problems can be avoided.
>
>
> 2) Role of priors is under-specified empirically: The paper sets
>  via a Student- kernel in the original space and discusses
> , but their effects are not isolated.
>
>  Answer: In the new version, we test the uniform distributions (See Table 9 in appendix).
>
>
> 3) Set-valued classification needs stronger baselines and clearer protocol. When set size is 1 and
> , the model degenerates toward a NN-style rule; comparisons to strong NN/prototypical baselines and train/test details are limited.
>
> Answer: In fact, it often uses some deep learning techniques to achieve good feature representations of data and then the nearest-neighbour classifier is used to classify the samples. If the size is 1, it is not necessary to use our method to classify  samples since we do not need to transport two groups of samples in such a case. Our method is meaningful for set-valued data.
>
>
> 4) No ablations for key hyperparameters: The objective exposes multiple hyperparameters
>  and design choices (e.g., neighborhood size
> ), but their effects are not isolated.
>
> Answer: In fact, we explore the effect of neighborhood size and hyperparameters in the old versions(see Figures 3, 5, 6 in the appendix). But these hypeparameters are often chosen by using  cross-validation on small and medium-sized data sets. On large-scale data sets, we should  first choose from  a small data set since directly training models is time-consuming.   In the new version, we add some experiments to show the effect of these hyperparmaeters.
>
>
> 5) Scalability reporting is missing: Although per-step complexities are stated, there is no wall-clock or memory usage across dataset sizes.
>
> Answer: In the old version. Fig.5 and Fig.6 show the running time of our model when hyperparameters change. In the new version, we fix some hyperparameters to test the running time of models. For memory usage, we compare some methods based on kernels with some methods based on deep-learning methods. For kernel-based methods, they store projection vectors. Our model needs to store the parameters of autoencoders.  Thus for autoencoders, the memory space depends on the structure of the neural network. For kernel methods, the memory depends on the number of projection vectors. Thus, autoencoders need a larger memory space than classical feature learning methods.
>
>
>
> 6) In Eq.(1), are marginals enforced hard via or soft via KL on marginals? If both are used, what is the KL term’s role? Please run an ablation: (a) hard-only (keep
>
> We correct the some mistakes for these formulas.
>
>
> 7) Please include runtime and peak-memory tables on small/medium/large datasets vs. baselines with identical hyperparameters.
>
>  The running time of our methods in  Figures 5, 6 in the appendix in the old version. We add the comparison of the running time in Table 8 in the appendix.For memory usage, we compare some methods based on kernels with some methods based on deep-learning methods. For kernel-based methods, they store projection vectors. Our model needs to store the parameters of autoencoders.  Thus for autoencoders, the memory space depends on the structure of the neural network. For kernel methods, the memory depends on the number of projection vectors. Thus, autoencoders need a larger memory space than classical feature learning methods.
>
> 8) How much performance comes from the priors
>  vs. the transport term? Please report an ablation (i) with priors and (ii)
> , including accuracy/NMI and sensitivity curves.
>
> Answer: Accuracy/NMI and sensitivity curves are found in Figures 5 and 6. and we make some experiments in terms of different hyperparameters(see Table 9 in the appendix).
>
> 9) Please add ablations for, neighborhood size
> , and the distance/order choice; report mean
>  CI over multiple seeds and include sensitivity curves.
>
> Answer:  The effect of neighborhood size and hyperparameters  is stated in the old version(see Figures 3 in the appendix),  and we report means and standard deviations of methods based on random experiments. Currently, the optimal transport is often used in terms of   the 2nd-order Wasserstein distance.  Our theoretical analysis depends on $Lp$ norm. Choosing the specific norm for a data set may be difficult since it is a hyperparameter. We only provide a framework of feature learning based on optimal transport and autoencoders. The choice of distance and autoencoders depends on users.

---

### Official Review · Reviewer_iUwh · 2025-11-02

**Soundness:** 2
**Presentation:** 2
**Contribution:** 2
**Rating:** 4
**Confidence:** 2

**Summary:**

The paper proposes a new feature learning framework combining Unbalanced Optimal Transport with Auto Encoder. The method uses prior label information to define a conditional distribution p_{j|i}, restricting transport to a local neighborhood of k similar samples. This mapping is implemented via an Auto-Encoder, and the overall optimization is solved using Block Coordinate Descent. Under unsupervised setting, the framework is extended by optimizing anchor points in feature space. Experiments on UCI and MNIST datasets demonstrate strong performance on supervised classification and unsupervised clustering tasks. The method also shows capabilities on set-valued classification tasks.

**Strengths:**

1. The idea of local matching is interesting for feature learning, and it verifies the effectiveness in supervised/unsupervised tasks.
2. Theoretically proves the generalization bound of the proposed method converges at 1/sqrt{n}.
3. Part of the optimization process has an analytical solution and is relatively efficient.

**Weaknesses:**

1. The convergence proof has not been carefully analyzed. The proof in [Razaviyayn,2012] only holds when each sub-problem is convex/quasi-convex, while Eq.(7) is not.
2. Calibration of the four regularized hyperparameters.
3. Related work about feature learning and the deep cluster method is not comprehensive.

**Questions:**

1. In the unsupervised setting, anchor points $z_j$ are learned in the feature space and updated each iteration. How is this fundamentally different from Deep Embedded Clustering, which is not discussed in the paper?
2. The algorithm has four hyperparameters. Is there a stable setting across tasks?

---

> ### Author Response · Authors · 2025-11-20
>
> Dear Reviewer iUwh
>
> We highly appreciate your time and effort to review our submission. We have uploaded a new version for our paper.
>
> 1)  The convergence proof has not been carefully analyzed. The proof in [Razaviyayn,2012] only holds when each sub-problem is convex/quasi-convex, while Eq.(7) is not.
> Answer: This may be one disadvantage of our method due to the use of the neural networks.  In classical optimization theory, some works have discussed the convergence of  alternating optimization.(  Convergence of alternating optimization, Bezdek, James C.  and  Hathaway, Richard J, Neural, Parallel and Scientific Computations, 2003). However, due to the use of deep neural networks, proving its convergence of our method may be challenging. Hence we remove the statement of the convergence of our model. We find that the trend of the value of the loss function decreases when  we directly use optimization tools.
>
> 2)  Calibration of the four regularized hyperparameters.
> We have made some experiments for these hyperparameters on Table 9 in the appendix.
>
> 3) Related work about feature learning and the deep cluster method is not comprehensive.
>
> Answer: Currently, numerous feature learning methods based on deep learning have been proposed. The deep clustering methods based on various neural networks have been devised. We only test the simple full-connected networks and do not consider the deep clustering methods based on complex neural networks. The aim of this paper is to provide a framework of learning features based on optimal transport and autoencoders.  Hence, we choose the relevant references for our work.
>
> 4) In the unsupervised setting, anchor points
>  are learned in the feature space and updated each iteration. How is this fundamentally different from Deep Embedded Clustering, which is not discussed in the paper?
>
>  Answer: Our model is also inspired from deep embedded clustering. Deep embedded clustering are designed by using other objective functions. Our method can adaptively learn the weights of samples in the  encoded space based on optimal transport. This may be a basic difference between our model and some deep embedded clustering methods. We mention these points in the new version.
>
>
> 5) The algorithm has four hyperparameters. Is there a stable setting across
>  tasks?
>
> Answer: Since we adopt the same encoders and decoders, a hyperparameter is set to zero.  Thus, there are three hyperparameters in our model. For small-scale data sets, we observe that the optimal hyperparameters vary as with change of data sets. On large-scale  data sets, they are relatively stable since there are many samples in the training set. On small-scale data sets, we often use cross-validation to choose these hyperparameters. For large-scale data sets, we choose these hyperparameters by making the experiments on a small training set. In the new version, we make some experiments to discuss the effect of the hyperparameters.

---

### Meta-Review · Area_Chair_194a · 2025-12-12

**Summary:**

This work proposes to learn features based on local unbalanced optimal transport and auto-encoders for classification/clustering. The strengths lie in the well-supported local-transport framework with a theoretical analysis. The weaknesses lie in the lack of a sound non-incremental technique, and the lack of proper positioning in the literature. Moreover, the reviewers have major concern on the convergence of the proposed algorithm. Based on the discussions and the rebuttals, the current work need more iterations and further development before getting published. I recommend rejection.

**Reviewer Concerns:**

The reviewers iUwh and xXdB are concerned on convergence of the block coordinate descent algorithm --- the authors acknowledged it as a disadvantage of the proposed method.

Most reviewers are concerned on the overall novelty of the method. After the rebuttal/revision, the paper still lacks comparison to SOTA baselines and proper positioning in the literature.

**Reviewer Scores:**

The original score is below the commonly perceived acceptance threshold. Based on the rebuttal, There is no clear evidence that the reviewers might increase the scores.

---

### Decision · Program_Chairs · 2026-01-26

Reject